# Biomolecular condensates form spatially inhomogeneous network fluids

Furqan Dar[1,8], Samuel R. Cohen[1,2,8], Diana M. Mitrea [3], Aaron H. Phillips[4], Gergely Nagy [5], Wellington C. Leite[5], Christopher B. Stanley [6], Jeong-Mo Choi [7,8] ✉, Richard W. Kriwacki [4] ✉ & Rohit V. Pappu [1] ✉

The functions of biomolecular condensates are thought to be influenced by their material properties, and these will be determined by the internal organization of molecules within condensates. However, structural characterizations of condensates are challenging, and rarely reported. Here, we deploy a combination of small angle neutron scattering, fluorescence recovery after photobleaching, and coarse-grained molecular dynamics simulations to provide structural descriptions of model condensates that are formed by macromolecules from nucleolar granular components (GCs). We show that these minimal facsimiles of GCs form condensates that are network fluids featuring spatial inhomogeneities across different length scales that reflect the contributions of distinct protein and peptide domains. The network-like inhomogeneous organization is characterized by a coexistence of liquid- and gas-like macromolecular densities that engenders bimodality of internal molecular dynamics. These insights suggest that condensates formed by multivalent proteins share features with network fluids formed by systems such as patchy or hairy colloids.

Biomolecular condensates are compositionally distinct membrane-less bodies that enable spatiotemporal organization and control over a range of biochemical reactions in cells[1–6]. Condensates are often thought of as spatially homogeneous liquids that form via liquid-liquid phase separation[7,8]. However, a more nuanced view is emerging. This is being driven by the realization of the importance of multivalence of associative motifs and domains as being crucial for driving condensation[9–11]. Reversible physical crosslinks formed among multivalent macromolecules, thus giving rise to networked molecules that underlie the viscoelasticity of condensates[12–21]. The phase transitions that give rise to condensates involve coupled associative and segregative phase transitions of associative macromolecules[22,23].

Proteins that are exemplars of associative macromolecules have distinct molecular features, typically encompassing oligomerization domains (OD), ligand binding domains, and intrinsically disordered regions (IDRs) with distinctive sequence characteristics[11,22,24–28]. The coupling of associative and segregative phase transitions, referred to as COAST[22], and the driving forces for these transitions derive from the molecular features of associative macromolecules[10,20,29–38]. Complex coacervation is a clear illustration of the coupling between associative and segregative phase transitions[11,28,39–45]. Here, the complexation of polyelectrolytes is driven by a combination of enthalpically favorable associations and entropically favored release of counterions[39,42,44,46]. If the complexes are higher-order clusters of undefined stoichiometry, then we arrive at the Ogston limit where size and hydration details

[1]Department of Biomedical Engineering and Center for Biomolecular Condensates, Washington University in St. Louis, St. Louis, MO 63130, USA. [2]Center of Regenerative Medicine, Washington University in St. Louis, St. Louis, MO 63130, USA. [3]Dewpoint Therapeutics Inc., 451 D Street, Boston, MA 02210, USA. [4]Department of Structural Biology, St. Jude Children's Research Hospital, Memphis, TN 38105, USA. [5]Neutron Scattering Division, Oak Ridge National Laboratory, Oak Ridge, TN 37831, USA. [6]Computational Sciences and Engineering Division, Oak Ridge National Laboratory, Oak Ridge, TN 37830, USA. [7]Department of Chemistry and Chemistry Institute for Functional Materials, Pusan National University, Busan 46241, Republic of Korea. [8]These authors contributed equally: Furqan Dar, Samuel R. Cohen, Jeong-Mo Choi. ✉e-mail: jmchoi@pusan.ac.kr; richard.kriwacki@stjude.org; pappu@wustl.edu

lower the solubility, and the complexes undergo a segregative transition to generate coexisting dilute and dense phases[22,47,48]. The dilute phase will comprise dissociated polyions, complexed polyions that form higher-order oligomers[49,50], and pre-percolation clusters[51,52], all of which must be electroneutral and hence will involve different degrees of ion associations. The dense phase will be a percolated network of polyions, whereby each polyion has multiple partners, and formally this will be limited by the number of uncompensated charges on each polyion in the dense phase. Accordingly, complex coacervation involves electrostatically-driven associations, which can give rise to higher-order complexes, and segregation into dense and dilute phases driven by a combination of counterion release and the lower solubility, due to altered hydration profiles, of higher-order complexes. Other instantiations of COAST-like processes include the coupling of percolation and phase separation[9,10,53,54]. Percolation, specifically bond percolation, also known as physical gelation[25,37,54,55], is a continuous associative phase transition whereby motifs or domains form reversible physical crosslinks to enable the formation of sequence- and architecture-specific networks that span the length scale of the system of interest[34,53]. As the networks grow, phase separation can be driven by the balance of inter-macromolecule, macromolecule-solvent, and solvent-solvent interactions, controlled largely by the sequence, structural, and solubility characteristics of spacers, which are regions outside the associative domains and motifs[47,56].

COAST-like processes give rise to condensates with network-like internal organization[30,57–59]. The networks will be defined by the architectures of the constituent molecules and the extent of crosslinking among the molecules[34,53,57,60]. The internal viscosity of condensates and the elasticity of the networks will be governed by the interplay between the timescales for molecular transport within and into/out of condensates and the timescales for making and breaking physical crosslinks. Furthermore, the network-like internal organization will engender spatial inhomogeneities of physically crosslinked macromolecules[22,29,30,58].

Viscoelastic materials have time-dependent properties and network structures contribute directly to viscoelastic moduli of condensates[61]. Even if condensates are dominantly viscous fluids as opposed to elastic solids, there will be timescales where the materials are dominantly elastic[58]. Condensates can age, and if they undergo equilibrium fluid-to-solid transitions, they transform from dominantly viscous to dominantly elastic viscoelastic materials[58]. Alternatively, some aged condensates can behave like viscoelastic network glasses[62,63]. While the network-like organization within condensates has been inferred from viscoelastic measurements and validated by the reproduction of measured moduli using computed network structures[58], there is a paucity of measurements that directly test the hypothesis of network structures within condensates.

The approach we pursue here is rooted in its historical use in the study of simple and complex fluids, and is based on scattering measurements, specifically small-angle neutron scattering (SANS)[64–67]. A key advantage of SANS is that one can investigate the presence of spatial inhomogeneities that range from a few angstroms to hundreds of nanometers[68]. Here, we investigate the structures of condensates that are mimics of nucleolar sub-phases. The nucleolus is a spatially organized condensate featuring at least three coexisting sub-phases. The GC, which is the outermost layer, is scaffolded by nucleophosmin (NPM1)[22,69,70]. Condensates formed by complexation of NPM1 and Arginine-rich (R-rich) peptides and proteins such as rpL5 and SURF6 have been used to test postulates of the molecular handoff model for ribosomal subunit assembly within the nucleolar GC[71–76]. Measurements of internal structure within condensates that are based on SANS were first reported by Mitrea et al.[71]. They studied condensates formed via heterotypic interactions of cationic arginine-rich peptides (rpL5) and N130, the N-terminal 130-residues of NPM1. The N130 construct includes the OD and at least three short regions that are rich in acidic residues[77].

Here, we revisit the SANS data collected by Mitrea et al.[71], updating these with new measurements and combining these with simulations to answer the following question: how might descriptors from theories of simple and complex fluids be adapted for describing condensates? To answer this question, we adapt approaches that integrate scattering data with computer simulations[78–85]. We combine traditional approaches based on pair distribution functions with graph-theoretic methods to arrive at descriptions of network structures of condensates formed by N130 and rpL5. The simulations we use are based on bespoke, sequence-specific coarse-grained (CG) models. The latter were developed using a machine-learning approach that is bootstrapped against atomistic simulations[86].

## Results

### N130 and rpL5 form condensates via complexation

Following the work of Mitrea et al.[71], the N130 construct corresponds to residues 1–130 of mouse NPM1 that includes the OD interspersed by short, disordered regions that encompass acidic residues (Fig. 1a). Previous work showed that N130 forms condensates with R-rich peptides. Two regions within N130 are enriched in acidic residues. One encompasses a flexible loop (residues 35–44, termed A1). The other is located at the C-terminus (residues 120–133, termed A2). These regions were shown to mediate interactions with R-rich peptides and promote condensate formation (Fig. 1a)[77]. There also is an N-terminal acidic region (residues 1–16, termed A0) that we discuss below.

We performed atomistic simulations using the ABSINTH implicit solvation model and forcefield paradigm[87]. In these simulations, the pentamerized OD was modeled as a rigid domain, the conformations adopted by the IDRs were sampled using Monte Carlo (MC) moves, and the simulations were performed at low salt concentrations of 20 mM. From the simulations, we obtained an overall structure of the N130 pentamer and an ensemble of conformations formed by N130 complexed rpL5 (Fig. 1b)[71]. The rpL5 peptide was taken from the ribosomal protein L5 and its sequence corresponds to the region that has been shown via experiments to interact with NPM1[77]. Simulations show that rpL5 adopts ensembles of expanded conformations that maximize the favorable solvation of Arg and Lys residues[88].

Titrating in rpL5 at a fixed N130 concentration of 100 μM in the absence of crowders gives a threshold rpL5 concentration for phase separation that is between 250 and 300 μM in 150 mM NaCl (Fig. 1c). The phase boundary (Fig. 1d) is consistent with previous experimental studies[71,89]. Next, we performed SANS measurements to probe the molecular organization within condensates formed by N130 complexed with rpL5 (Fig. 1e). The SANS intensity is a convolution of the form factor and structure factor. The former quantifies scattering that results from the average shapes of the scatterers, whereas the latter quantifies how the particles scatter neutrons due to spatial correlations caused by intra- and intermolecular interactions. Specifically, the structure factor measures density correlations in reciprocal space, whereas the form factor is the Fourier transform of the density distribution[90].

The importance of complexation as a driver of internal organization is made clear by the lack of peaks in the scattering profile for N130 pentamers when rpL5 is absent from the solution (also see Supplementary Fig. 1). The inhomogeneities in spatial densities that are evident in the scattering profile (shown by the arrows of Fig. 1e) are indicative of order on specific length scales. The multi-peak fitting analysis, developed by Mitrea et al.[71], combined with analysis of derivatives of the scattering profile show that the most reliable, high signal-to-noise peaks correspond to length scales of ~55 Å and ~77 Å (Fig. 1e). In the derivative analysis, the signal-to-noise is found to decrease at low $q$ values, and this makes the unambiguous assignments of peaks beyond the second one more unreliable (Supplementary Fig. 2).

The molecular diameter of the pentamerized OD (~53 Å) is a useful ruler for calibrating the different length scales. To further characterize

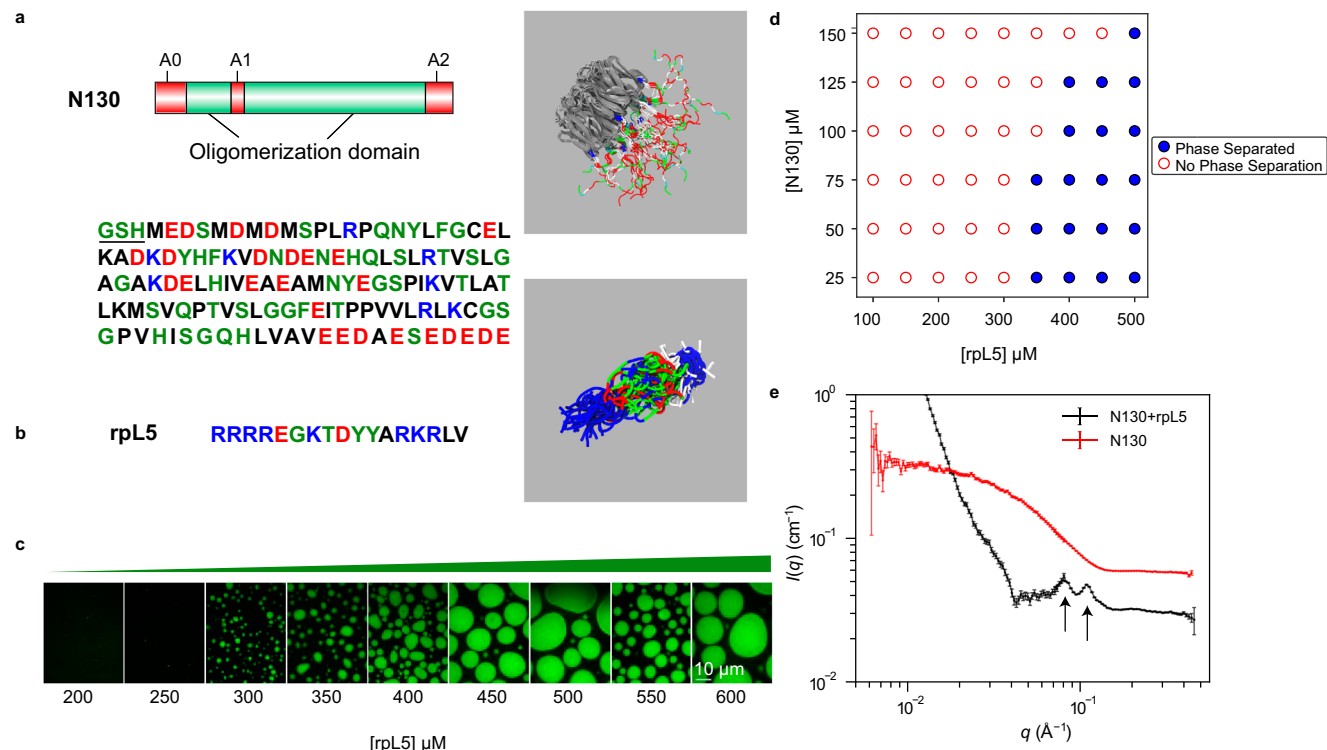

**Fig. 1 | Complexation between acidic regions within N130 and R-motifs of rpL5 is required for condensation. a** Schematic representation of N130 including the different acidic regions. The amino acid sequence of N130 is also shown. The three acidic regions A0, A1, and A2 span residues 4–18, 35–44, and 120–133, respectively. Non-native N-terminal residues that remain after protease cleavage are underlined. On the right, we show the overall structure of the hairy colloid generated by superposition of 50 distinct conformations from atomistic simulations. These simulations use the ABSINTH implicit solvent model[87] and explicit representations of solution ions[132] (which are not shown in the figure). The pentamerized OD (PDB ID 4N8M), in gray, was modeled as a rigid molecule in the atomistic simulations. **b** The amino acid sequence of rpL5. The panel on the right shows a superposition of 50 different conformations extracted from ABSINTH-based simulations. **c** Confocal microscopy images of phase separation of 100 μM N130 upon titrating the concentration of rpL5 in buffer and 150 mM NaCl. N130 is labeled with AlexaFluor488. Microscopy experiments were repeated in three independent experiments with

similar results. **d** Two-component phase boundary for N130 + rpL5, showing the result of concentration titrations. **e** SANS curve showing the intensity $I(q)$ plotted against $q$, the scattering vector, for condensates formed by a solution of N130 (200 μM): rpL5 at 1:3 stoichiometry. Multi-peak analysis, as described by Mitrea et al.[71], leads to the identification of two major peaks corresponding to 55 Å (right arrow) and 77 Å (left arrow) that are annotated on the figure. The SANS curve for N130 pentamers, in the absence of rpL5, is shown for comparison. In the interest of clarity, this curve is shifted upwards vis-à-vis the curve for the N130 + rpL5 system. We computed the scattering curve for individual N130 pentamers. These computed profiles show qualitative resemblance to the SANS curve shown here for N130 pentamers (see Supplementary Fig. 1). The error bars are propagated through each of the SANS data reduction steps, described and cited in the Methods. Intensity data are binned and circularly averaged to convert the 2D detector data into $I(q)$ data, from which we calculate uncertainty values at each value of $q$. The data correspond to a single measurement ($n = 1$).

the nature of the ordering and the interactions that contribute to ordering, we turned to computational approaches.

## Systematic CG and predictions

To investigate the internal structure of fluid-like condensates, we performed CG simulations of the N130 + rpL5 condensates. In the CG model, the pentamerized OD of the N130 pentamer, referred to hereafter as PD, was modeled as a single, spherical bead defined by excluded-volume interactions. We used a single-bead-per-residue representation for residues in the IDRs of N130. Accordingly, in addition to the acidic regions, A1 and A2, we also explicitly modeled the N-terminal region of N130 (termed A0). The architectures of the CG N130 molecules are reminiscent of hairy colloids[91], featuring disordered, acidic regions that protrude from one side of the sphere that mimics the PD. Hairy colloids are known to form network fluids through anisotropic interactions engendered by the architectures of the constituent molecules[92–96]. All residues in rpL5 were modeled as single beads.

The systematic CG procedure was initiated by bootstrapping against information generated using atomistic simulations based on the ABSINTH implicit solvation model and forcefield paradigm[87] (Fig. 2a). Having prescribed the resolution for the CG model, we then

used ensembles from atomistic simulations of N130 pentamers with 15 copies of rpL5 to generate forcefield parameters for the CG model. For this, we use the CAMELOT algorithm[30,86,97] that combines a Gaussian Process Bayesian Optimization[98] module, with an appropriate architecture and CG model. The parameters of the CG model minimize the difference between the atomistic conformational ensembles and the CG representation. This affords the dual advantages of computational efficiency afforded by the CG model and the sequence-specific effects learned via the CAMELOT algorithm. Using the CG representation, we simulate a dense phase with 108 copies of N130 and 1620 copies of the rpL5 peptide.

Results from the CG simulations of dense phases were used to compute inter-residue contact maps between the disordered regions of N130 and the rpL5 peptide (Fig. 2b). The A1 and A2 regions make favorable contacts with the basic residues in rpL5. We also observed that the A0 region makes contacts with the basic residues in rpL5. The frequency of contacts suggests that this region forms stronger interactions with rpL5 than A1. The contacts involve acidic residues within A0. The contact maps derived from the CG simulations suggest a rank ordering of interactions between acidic regions and rpL5, with A2 being the most favorable and A1 the least.

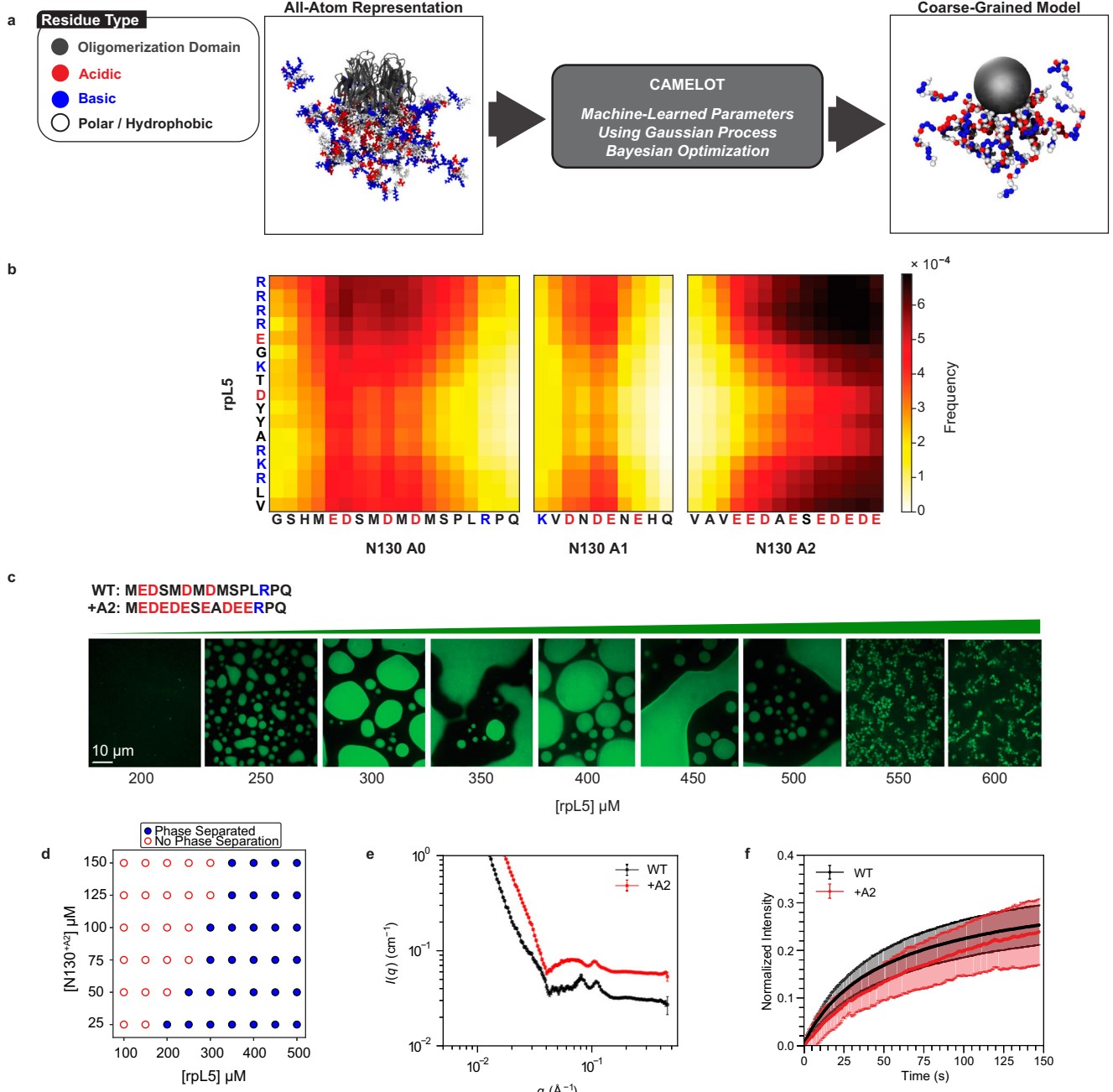

**Fig. 2 | Coarse-grained simulations of N130 + rpL5 condensates highlight the importance of an N-terminal acidic region (A0) within N130. a** Coarse-graining procedure for N130 and rpL5 systems. To generate sequence- and system-specific CG models for N130 and rpL5 peptides, we start with atomistic Monte Carlo simulations of individual molecules that are based on the ABSINTH implicit solvation model and forcefield paradigm[87]. Ions are modeled explicitly in these simulations. We then prescribe a CG model for the system. Here, the residues of the PD are collectively modeled as one large bead depicted in gray. Next, for regions outside the PD, the residues are single beads, and the bead types are organized into three groups: 1: E,K,R,D; 2: V,F,M,I,L,Y,Q,N,W,H; and 3: A,P,G,S,T,C, respectively. To determine the optimal interaction parameters for the CG model, we used the CAMELOT algorithm[86]. **b** Bead-to-bead contact maps from coarse-grained simulations of dense phases comprising a 1:3 ratio of N130 and rpL5. The A1 and A2 regions

interact favorably with rpL5 peptides. The contact maps reveal interactions involving a region we refer to as A0. **c** Confocal microscopy images of phase separation at a fixed concentration of N130$^{+A2}$ upon titrating with rpL5. The A0 tract in the wild type is replaced with the reversed sequence of the A2 tract. Microscopy experiments were repeated in three independent experiments with similar results. **d** Two-component phase boundary for N130$^{+A2}$ + rpL5, showing the result of concentration titrations. **e** Comparative SANS curves for the WT N130 + rpL5 and +A2 mutant + rpL5. SANS data were collected with 200 μM N130 and 600 μM rpL5. For clarity, the curve for N130$^{+A2}$ + rpL5 is shifted upwards. The data correspond to a single measurement ($n = 1$), with the error bars calculated as in Fig. 1e. **f** FRAP curves for the condensates containing N130 and N130$^{+A2}$. FRAP curves were collected at 100 μM N130 and 400 μM rpL5 with 12 replicates ($n = 12$). Error bars represent the standard deviation of the mean.

Our predictions motivated the generation of a new mutant construct where we replaced A0 with the residues from A2, in reverse order, to increase the linear charge density (see the sequence of the new A0 region in Fig. 2c). We refer to this construct as N130$^{+A2}$. It has

more acidic residues than the wild type. We hypothesized, based on simulations, that the +A2 mutant should form condensates with a lower threshold concentration of rpL5 for a given N130 concentration. Indeed, titrating in rpL5 at a fixed concentration of N130$^{+A2}$ leads to a

lower rpL5 threshold concentration (Fig. 2d) when compared to the threshold concentration that is required for condensation with wild-type N130 (Fig. 1c). Increasing the strength of the electrostatic interactions in A0 reduces the threshold concentration for rpL5 from 400 μM to below 350 μM at 100 μM N130. Note that the designs were chosen to ensure that the stoichiometric ratio required for condensation does not change.

Next, we investigated the impact of the +A2 mutant using SANS (Fig. 2e). We observed similar pairs of peaks at intermediate $q$-values for both N130 + rpL5 and the +A2 mutant + rpL5. Small shifts in the locations of the peaks are likely a combination of inherent noise and a contribution from electrostatic repulsions in the disordered N- and C-termini of N130 emanating from the same face of the PD[77]. The C-terminus of the wild-type protein contains nine negatively charged residues corresponding to A2, and the +A2 mutant increases the net charge on the pentamer by 25.

We also measured the impact of the +A2 mutant on the internal dynamics of N130 + rpL5. For this, we performed measurements of fluorescence recovery after photobleaching (FRAP) on the condensates (Fig. 2f). The FRAP curve for N130 + rpL5 indicates dynamical exchange with the bulk solution with the recovery time constant being 53 ± 2 s. Increasing the total charge on N130 via the +A2 mutant decreases the overall extent of FRAP, resulting in a longer recovery time of 103 ± 8 s. Similarly, we observe that N130$^{+A2}$ + rpL5 displays slower overall dynamics at shorter timescales, and the dynamics of the two systems approach one another at longer times. The average recovery times were obtained by fitting the data, for both constructs, to a single species model. This ignores the prospect of there being an immobile fraction. However, since FRAP data are a convolution of contributions from physical crosslinks and molecular transport, we chose a parsimonious, single-species model to avoid over-fitting and over-interpretations of the data.

### Condensates formed by N130 + rpL5 are network fluids

As observed in the SANS data (Fig. 1e), N130 + rpL5 condensates display correlations at length scales that are consistent with dimensions of the PD of N130. Therefore, we focus our analysis on the spatial correlations formed by N130 within condensates. Obtaining the experimental

structure factor by deconvolution of the SANS spectrum would require modeling the form factor. This becomes intractable given the geometry of the molecules[99]. Instead of solving an inverse problem, we computed pairwise correlations via the radial distribution function (RDF) $g(r)$. This is the real-space analog of the experimentally measured structure factor[100]. It describes how spatial densities change as a function of distance from an arbitrary reference particle. Normalized to an ideal gas, where the distance between particle pairs is completely uncorrelated, $g(r)$ is the standard descriptor of liquid structure in theory, experiment, and simulations.

There are accounts of condensates being akin to simple liquids[7]. However, in the physical literature, the term "simple liquids" refers to fluids formed by Lennard-Jones (LJ) particles. Accordingly, we calibrated our expectations regarding the organization of N130 and rpL5 molecules within condensates, using the RDF, $g(r)$, for the LJ fluid as a touchstone (Supplementary Fig. 3). In an LJ fluid, structure is defined purely by packing considerations[101].

In any $g(r)$, the first peak corresponds to the nearest neighbors in the vicinity of the reference particle of diameter σ, and the additional peaks correspond to higher-order neighbors in surrounding shells. As a measure of the density correlations, $g(r)$ quantifies how the average density at a separation $r$ from the center of any particle varies with respect to the average density of the fluid. The density correlations are large in the vicinity of the reference particle, and the relative probability, vis-à-vis the ideal gas, decays as a function of distance until the density becomes indistinguishable from the average density of the fluid[102]. Structure can be further characterized by the volume integral over $g(r)$ up to defined positions such as the first minimum. This quantifies the nearest-neighbor coordination number[100]. For the LJ fluid, the coordination number is ~12–13 due to optimal packing of the spherical particles. In contrast, complex fluids have open, network-like organization due in part to less efficient packing.

From the CG simulations, we computed $g_{PD-PD}(r)$, which quantifies spatial correlations between pairs of PDs of different N130 molecules (Fig. 3a). The profile for $g_{PD-PD}(r)$ is consistent with liquid-like organization, featuring short-range order and long-range disorder with $g_{PD-PD}(r)$ approaching unity at large distances. Here, disorder, which refers to the length scale at which $g_{PD-PD}(r)$ approaches unity, is evident

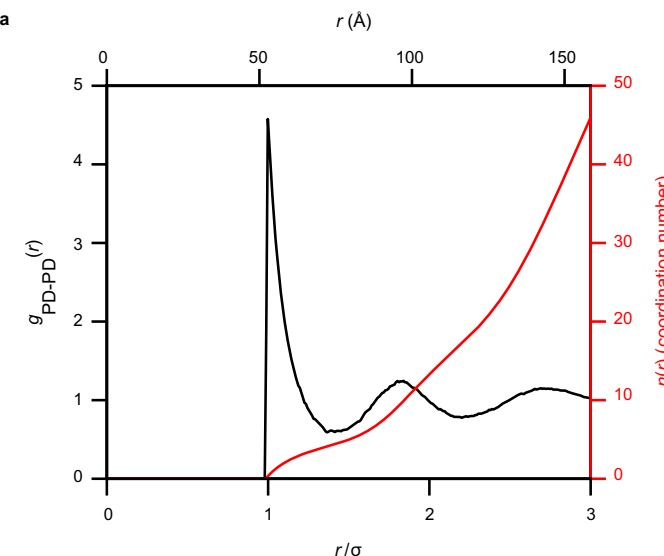
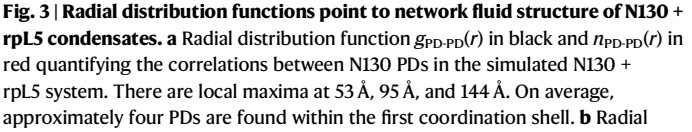
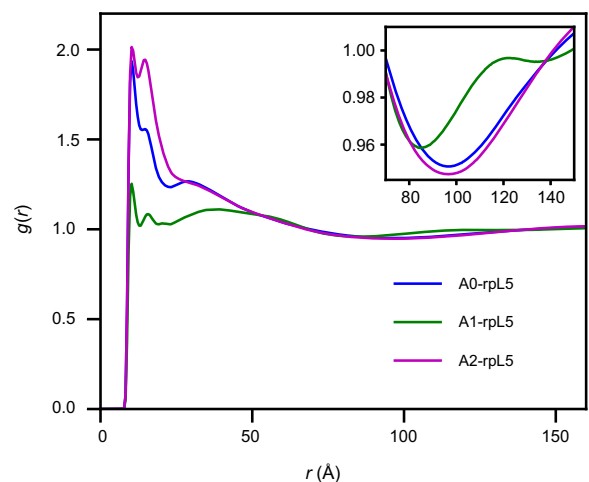

**Fig. 3 | Radial distribution functions point to network fluid structure of N130 + rpL5 condensates. a** Radial distribution function $g_{PD-PD}(r)$ in black and $n_{PD-PD}(r)$ in red quantifying the correlations between N130 PDs in the simulated N130 + rpL5 system. There are local maxima at 53 Å, 95 Å, and 144 Å. On average, approximately four PDs are found within the first coordination shell. **b** Radial distribution functions $g_{+-}(r)$ quantifying the correlations between positive charges in rpL5 and negative charges in acidic regions of N130. The inset shows a zoom in to troughs in $g(r)$ in a spatial range between 75 Å and 140 Å that is between 1.5σ and 2.5σ, where σ is the diameter of the N130 PD. Averages were calculated over five replicates ($n = 5$).

beyond an inter-PD distance of $3\sigma$, where $\sigma \approx 53$ Å is the diameter of an N130 pentamer. Integration of $g_{PD\text{-}PD}(r)$ up to the first minimum yields a coordination number of approximately four. This suggests that the average structure of the fluid, as interrogated from the vantage point of N130, is not determined purely by packing considerations, as would be the case for an LJ fluid. Instead, rather like liquid water, which has a coordination number of approximately four, defined by networks of hydrogen bonds, the N130 molecules makeup a network fluid.

The peaks in $g_{PD\text{-}PD}(r)$ occur at 53 Å, 95 Å, and 144 Å. The second and third peaks correspond to ordering beyond the molecular length scale. The ratios of the computed peaks to those estimated based on SANS measurements are 0.96 and 1.25 for the first and second, peaks, respectively. Note that the estimates of higher-order peaks from SANS data are less reliable given lower signal-to-noise as quantified using analysis of the derivatives (Supplementary Fig. 2). Further, the parameters of the CG model, especially the parameters for Van der Waals interactions, which are governed by the inter-residue and inter-domain distances, will depend on the screening length and ion-mediated correlations in atomistic simulations. The ABSINTH model includes explicit representations of solution ions, and these simulations were performed at low ionic strength, with the salt concentration set at 20 mM given the explicit representations of ions and large droplet sizes. The inclusion of explicit representations of ions leads to exponential increases in simulation time because of the way electrostatic interactions are handled in the ABSINTH model[87]. In the SANS measurements, the salt concentrations were 150 mM. Therefore, given the parameterization of the CG model using atomistic simulations, the differences in peak positions that correspond to intermediate and longer-range ordering are due to differences in effective Debye lengths between the simulations and SANS measurements. Because the Van der Waals parameters are learned from atomistic simulations, one cannot achieve perfect congruence by simply changing Debye lengths in the CG simulations. Instead, we need salt concentration dependent parameters within the CG model. This requires a model for how the salt-dependent interactions change at different length scales. The remainder of the discussion focuses on insights we can glean from the CG simulations. In doing so, we presume semi-quantitative congruence with SANS experiments.

Next, we computed the $g(r)$ between basic residues in rpL5 and acidic residues in the three different regions of N130 (Fig. 3b). Note that these $g(r)$ profiles were computed as a linear superposition of pair distributions between all the basic residues in the peptide and all the acidic residues in a specific region. Each of these $g(r)$ profiles has distinct peak positions and heights for the first maximum. The heights of the peaks, realized in a range of $r < 50$ Å, are highest for A2 and lowest for A1. These trends are observed in the corresponding potentials of mean force (Supplementary Fig. 4). The most favorable interactions in the distance range of $r < 50$ Å are realized between A2 and rpL5. The hierarchy of interactions encoded in the different acidic regions of N130 agrees with the contact maps (Fig. 2b).

## Interactions between acidic regions and rpL5
Next, we investigated the effects of in silico mutations where we neutralized the charges within each of acidic region while keeping the density of the simulated dense phases fixed to that of the wild type N130 + rpL5 condensates. These simulations were designed to assess how mutations that affect electrostatic interactions mediated by one region affect the totality of the network structure. We computed $g(r)$ between pairs of PDs and between the acidic regions on N130 and basic residues of rpL5. Neutralizing the acidic residues on any of the three regions leads to a reduction in the first maximum of $g_{PD\text{-}PD}(r)$ (Fig. 4a). The magnitude of the reduction in the first maximum is greatest for the A2 mutant, followed by the A0 and A1 mutants, for which the values of the peak heights are statistically similar within error (Supplementary Fig. 5). This indicates that mutations to A2 affect the overall structure

more than mutations to the other regions. The potentials of mean force (Supplementary Fig. 6) corroborate this inference showing that the least favorable interactions involve the N130 PD of the A2 mutant. The interactions between basic residues of rpL5 and acidic residues within A0, A1, and A2 are modular. This is clear from the $g_{AX\text{-}rpL5}(r)$ profiles (Fig. 4b–d), where $X = 0$, 1, or 2, which we compute from simulations where one of A0, A1, or A2 is neutralized. The $g_{AX\text{-}rpL5}(r)$ profile deviates from that of the WT only for the region in which the charges are neutralized. Otherwise, the profiles remain roughly equivalent to those obtained from the wild-type N130 + rpL5 condensates. This suggests that the acidic regions make modular, and seemingly independent interactions with rpL5 peptides (Fig. 4b–d).

## Graph-theoretic analyses of network structures of condensates
The simulation results suggest that N130 + rpL5 condensates are network fluids as opposed to simple liquids. To put the network fluid concept on a quantitative footing, we turned to graph-theoretic analysis. These approaches have been used to analyze network fluids such as hydrogen-bonding networks[103–108] and network glasses[109].

Adapting precedents from work on molecular fluids[110], we construct unweighted graphs in which two molecules are considered adjacent if any of the constituent beads are within the cutoff distance defined by the first minimum in the corresponding $g(r)$. Using this criterion, we constructed adjacency matrices via block summations (Fig. 5). We then analyzed the network structure formed by the molecular neighbors for the set of beads considered.

To provide a suitable prior of a non-networked fluid where structure is dominated by packing considerations alone, we performed graph-theoretic analyses on systems of LJ particles. For this, we quantified the degree distributions for the vapor, liquid, and solid phases of spherical particles interacting via LJ potentials (Fig. 6a). The degree reflects the number of connections or edges emanating from a node. Here, a node is an individual LJ particle. For an ideal gas, the degree is zero. However, since LJ particles have finite size and there are attractive dispersion interactions, the vapor phase is not ideal. Instead, the degree distribution is skewed to the right. For the LJ liquid, we observe a broad distribution that is roughly symmetrical about a mean degree value of 13. As the density is further increased to obtain a solid, we see that the degree distribution shows a sharp peak at twelve, corresponding to the number of neighbors expected for a 3D hexagonal close-packed lattice[111]. The locally inhomogeneous nature of a liquid allows for interactions with more neighbors than the true ground-state number seen in the solid phase.

Next, we constructed graphs using acidic residues in N130 and the basic residues in rpL5 as nodes. In contrast to the LJ systems, the computed degree distributions are bimodal (Fig. 6b), and this is suggestive of a bipartite network structure. The multimeric nature of the N130 pentamer allows the acidic regions to interact with multiple rpL5 peptides, as seen in the broad second peaks in the degree distributions. Consistent with the RDFs for N130 + rpL5, we also observe a hierarchy of degrees, with A2-rpL5 having the largest degree and A1-rpL5 having the smallest. However, for the first peaks near $k = 0$, which correspond to the smaller rpL5, we see that the different acidic regions do not show appreciable differences. Comparison to the LJ system suggests that the N130 + rpL5 system features both liquid- and gas-like interactions. Here, the term "gas" refers to the presence of unassociated, freely diffusing rpL5 molecules that coexist with a liquid comprising associated rpL5 molecules.

## Dynamics within network fluids show two distinct regimes
In spatially inhomogeneous systems, there can be regions that are locally dense or dilute. This is made clear in the graph-theoretic analysis, which shows two interaction modalities. Similar results have been reported for fluids formed by patchy particles, especially near the liquid-gas coexistence region[92–94]. We reasoned that the coexistence of

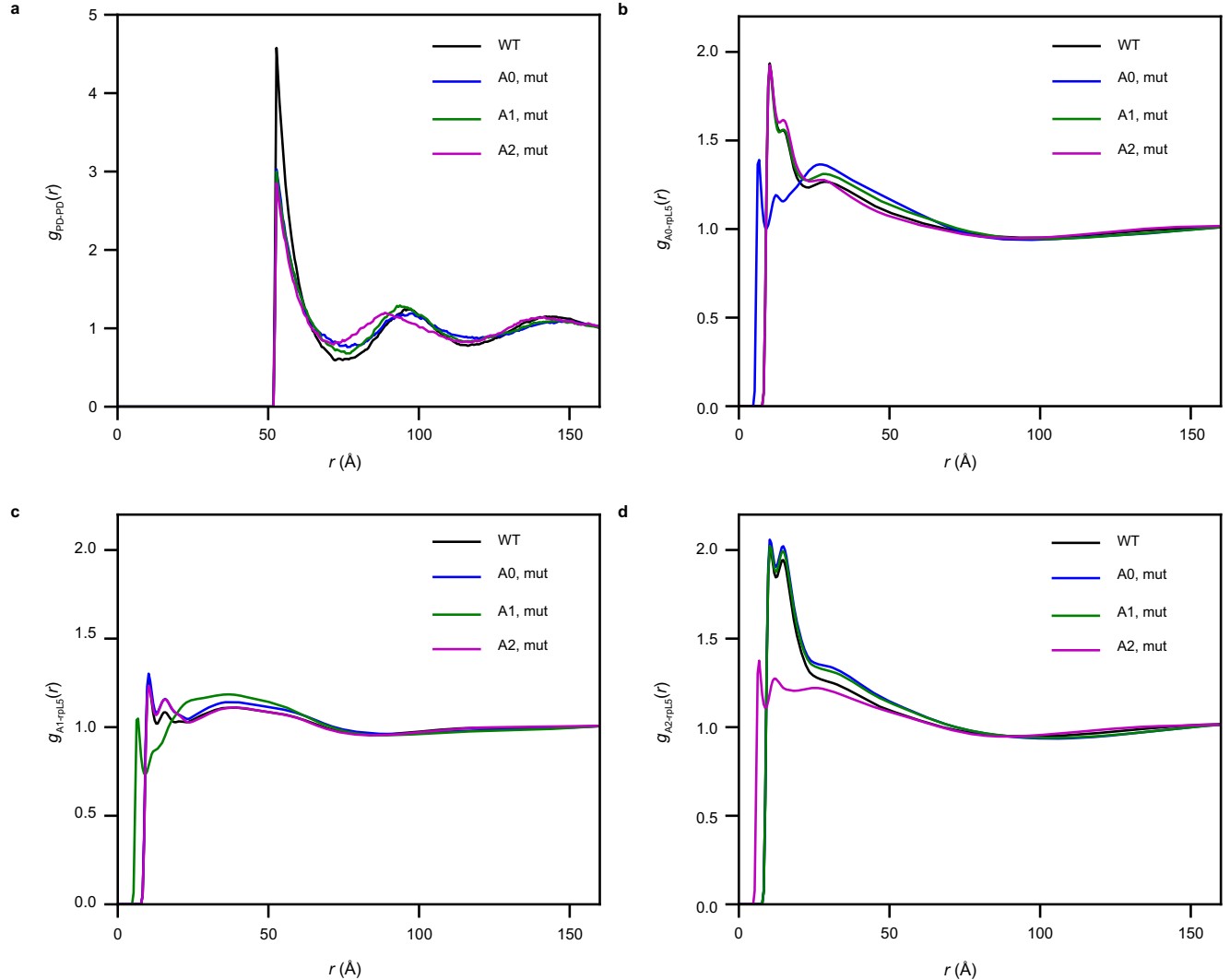

**Fig. 4 | Interactions between acidic regions and rpL5 are modular and independent of one another. a** $g_{PD-PD}(r)$ computed after neutralizing the charges in each of the acidic regions indicated in the legend. **b** $g_{A0-rpL5}(r)$ quantifies the pair correlations between acidic residues in A0 and basic residues in rpL5. Results are shown for the WT (black), when acidic residues are neutralized in A0 (blue), A1 (green), and A2 (magenta). Neutralizing charges within A0 weakens the interactions between A0 and rpL5. However, neutralizing the charges within A1 and A2 does not significantly affect the interactions between A0 and rpL5. **c** $g_{A0-rpL5}(r)$ quantifies the pair correlations between acidic residues in A1 and basic residues in rpL5. Results are shown for the WT (black), when acidic residues are neutralized in A0 (blue), A1 (green), and A2 (magenta). Neutralizing charges within A1 weakens the interactions between A1 and rpL5 (green curve). However, neutralizing the charges within A0 and A2 does not significantly affect the interactions between A1 and rpL5. **d** $g_{A2-rpL5}(r)$ quantifies the pair correlations between acidic residues in A2 and basic residues in rpL5. Results are shown for the WT (black), when acidic residues are neutralized in A0 (blue), A1 (green), and A2 (magenta). Neutralizing charges within A2 weakens the interactions between A2 and rpL5 (magenta curve). However, neutralizing the charges within A0 and A1 does not significantly affect the interactions between A2 and rpL5. In all cases, averages were calculated over five replicates ($n = 5$).

liquid- and gas-like organization within the condensates should have dynamical consequences. To test for this, we analyzed the simulations to compute mean square displacements (MSDs) of the PDs. The MSD is calculated as a function of lag time. This involves a double average, where the inner average is a cumulative sum along the time axis, starting from zero, and progressing in increments of $t + \Delta$, where the MSD is computed over times $t$ and $t + \Delta$ and averaged over the motions of individual molecules. The outer average is over all molecules. A characteristic timescale corresponds to the time it takes for the PD to diffuse across a distance corresponding to its diameter. We rescaled the abscissa by $t_D$, which is the timescale over which the motion of the PD fits best to a purely diffusive model with MSD being proportional to $t$. We find that there is a timescale below $t_D$ where the motion is super-diffusive with an exponent greater than one, and a timescale above $t_D$ where the motion of the PD is sub-diffusive with an exponent less than

one (Fig. 7a). Based on the observed length scales, the super-diffusive motion reflects the contributions of short-range steric repulsions among the PDs and the electrostatic repulsions between acidic residues. Conversely, the sub-diffusive motions reflect contributions from physical crosslinks between acidic residues and rpL5 peptides. Histograms of the exponents that we compute for the MSDs show a bimodal distribution (Fig. 7b). The distribution of sub-diffusive exponents is broader and reflects the heterogeneities of motions impacted by associative interactions between acidic regions and rpL5 peptides. The MSDs calculated for the PD and for charged residues in each of the acidic regions and the basic residues in the rpL5 peptides contrast with the MSDs computed in terms of the PD alone (Fig. 7c). The acidic regions and the peptides show sub-diffusive motions on all timescales, reflecting the fact that these moieties are influenced mainly by associative intermolecular interactions.

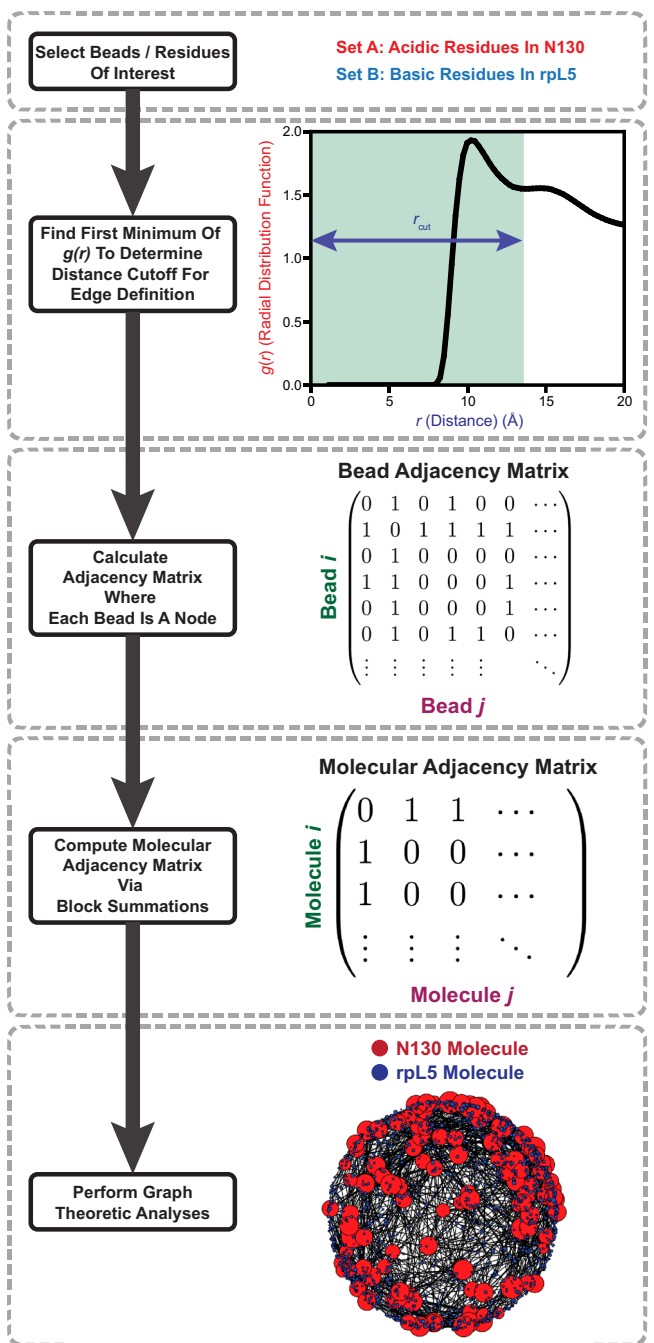

**Fig. 5 | Flowchart describing the graph-theoretic analysis of simulations of dense phases of N130 and rpL5.** We start by selecting a set of residues of interest. As an illustrative example, we pick the acidic residues in N130 and the basic residues in rpL5. To determine an edge, we compute the $g(r)$ between sets of beads where the first minimum serves as the distance cutoff for bead adjacency. Given this cutoff, we construct the bead-to-bead adjacency matrices. The system includes the N130 PD. Performing appropriate block summations of the bead-to-bead adjacency matrix generates the molecular adjacency matrix. The latter is analyzed using standard graph-theoretic analyses. As an example, a random embedding is shown where the size of a node corresponds to the degree of centrality of that node.

## Discussion

Condensates have been referred to as simple liquids[7] or as structureless entities characterized by non-specific interactions[112]. Systems of LJ particles form simple liquids, and macromolecules that drive condensation are not LJ particles. Further, liquids are not structureless entities. *Ad hoc* criteria are often used to define liquids in the

condensate literature[8]. Instead, structure in liquids is characterized by short-range order and long-range disorder. The order parameter for describing liquid-state structure is the RDF. The extent and range of orders that can be quantified using RDFs are directly connected to the molecular architectures, the spatial range, types, and strengths of intermolecular interactions[102]. SANS measurements are particularly useful for gleaning quantitative insights regarding RDFs.

Here, we deployed a combination of experimental and computational techniques to demonstrate that condensates formed by N130 and rpL5 are network fluids. This was established by observing peaks in SANS curves of condensates that are indicative of molecular order on the length scale of the N130 pentamers. The SANS data and computations show short- and intermediate-range ordering versus long-range disorder. From the computed $g_{PD-PD}(r)$ profiles, we find that N130 pentamers have four nearest neighbors on average. Complexation between the acidic regions and the rpL5 peptides also contributes to the overall structure of the condensates. The acidic regions of N130 function as independent interaction modules. This explains why the valence of cohesive motifs is an important driver of condensation and material properties[34–37,71].

The sequence-specific CG model allowed us to identify a new acidic region, termed A0, in the N-terminal end of N130. We find that A0 interacts more strongly with the disordered peptide rpL5 than was hitherto appreciated. Mutations that increase the charge in A0 help lower the threshold concentration of rpL5 that is needed to observe condensation driven by heterotypic interactions with N130.

We find that there are two types of sub-graphs that underlie the structure of the N130 + rpL5 condensates. One of the sub-graphs corresponds to gas-like organization, and the other corresponds to that of a liquid. Note that "gas-like" implies that there are regions within condensates where the concentrations of macromolecules are ultra-dilute, and hence solvent filled. This is akin to the empty liquid concept[113] reported for patchy colloids. Conversely, what we refer to as "liquid-like" refers to regions that are dense in macromolecules. The bipartite graphs also have dynamical fingerprints, which are manifest as the bimodality we observe for the MSDs of the PDs. Super- and sub-diffusive behaviors that we report here have been observed in MSDs computed from simulations of oligomer-grafted nanoparticles[114]. They are also consistent with data from nuclear magnetic resonance experiments where Gibbs et al. found that the PDs of NPM1 form an immobilized scaffold in NPM1 + p14ARF mixtures[115]. Taken together, our findings place the N130 + rpL5 system, and other such systems, in the same category as patchy and/or hairy colloids[92–94,96,113,114,116,117].

Here, we focused mainly on the effects of heterotypic interactions between N130 and R-rich rpL5 peptides on the network structure of the N130 + rpL5 condensates. Previous work has shown that the homotypic interactions within NPM1, uncovered in the presence of crowders, can also affect both the phase behavior and the mesoscopic structure of condensates formed with SURF6N[73,74]. A new method that leveraged the Edmond-Ogston formalism[118], allows for the intrinsic strengths of homotypic interactions to be uncovered using crowder titrations[48]. Knowledge of the strengths of homotypic interactions, and the relative interplay with heterotypic interactions, will allow for simulations of binary and higher-order mixtures that mimic nucleolar GCs. An application of graph-theoretic analysis, guided by SANS measurements, to condensates that form under the competing interplay of homotypic and heterotypic interactions should be feasible. The interplay between network structures defined by the whole range of homotypic and heterotypic interactions should illuminate the relationship between rheological properties and network structure[58], for mapping intra-condensate spatial organizational preferences[29], and for dynamical control over compositional identities of protein-RNA condensates[28].

Since the nucleolus is a multicomponent and multiphasic condensate[70], we expect that varying the stoichiometries of different

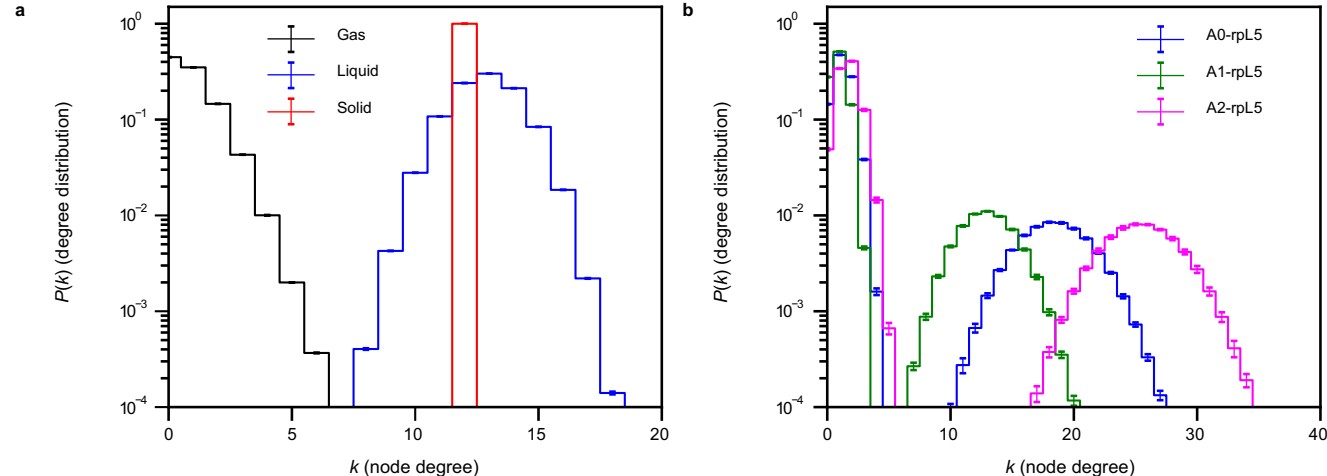

**Fig. 6 | Each acidic region of N130 in the N130 + rpL5 dense phase imparts a different network structure onto the system. a** Degree distributions, $P(k)$, for pure LJ systems. Because the density of the vapor ($\rho = 0.01, T = 1.000$) is low, the most likely degree is zero. For the liquid ($\rho = 0.80, T = 1.000$), the degree distribution shifts towards higher values with a mean around twelve, where the widths of the distribution correspond to the inherent variation in the number of bonds that particles can make in the locally spatially inhomogeneous environment of a liquid. For the solid ($\rho = 1.5, T = 0.758$), the degree distribution is peaked at twelve. Note

that we use reduced units for density and temperature. **b** Degree distributions, $P(k)$, for the complementary charge interactions between the different acidic regions of N130 and the rpL5 peptides. Unlike the graphs in **a**, the distributions display bimodality, which is an indication of a bipartite graph. Consistent with the radial distribution functions in Fig. 3, we see that A2 has the largest degrees, followed by A0 and A1. In all cases, averages were calculated across five replicates ($n = 5$).

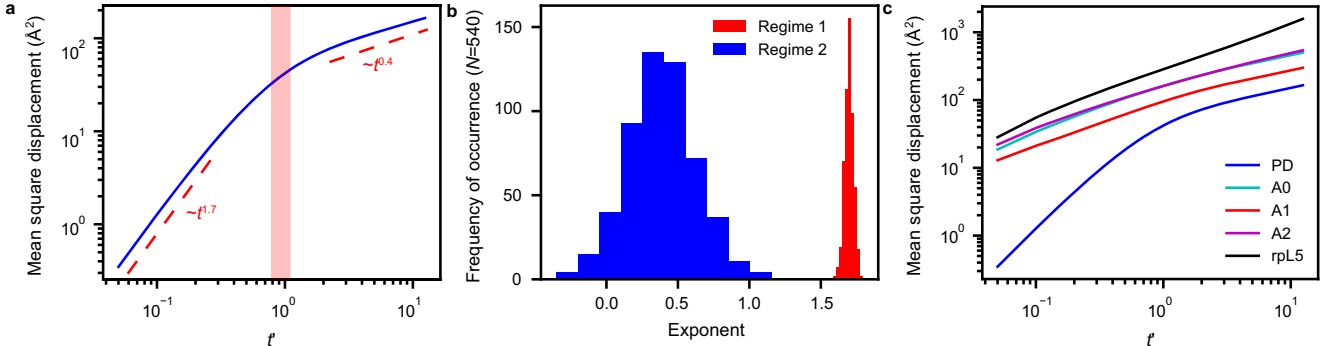

**Fig. 7 | Motions within dense phases of N130 + rpL5 show bimodality. a** MSD of the PD plotted against the lag time shows that N130 has super-diffusive and sub-diffusive regimes. Here, the abscissa is a unitless parameter $t' = t/t_D$ where $t_D$ is the timescale over which the motion of the PD is purely diffusive. The red region in the panel indicates the timescales that fit best to purely diffusive motion. There is a timescale below $t_D$ where the motion is super-diffusive and a timescale above $t_D$ where the motion is sub-diffusive, with the regimes and corresponding exponents

indicated in the panel. **b** Histograms of the exponents calculated for the mean square displacements of individual PDs. The bimodal distribution reflects the presence of super-diffusive and sub-diffusive regimes. Negative exponents result from uncertainty regarding the long-time behavior of the MSDs. **c** MSDs of the PD and charged residues in each of the acidic regions and the rpL5 peptides. In contrast to the PD, the acidic regions and the peptides show only sub-diffusive motions on all timescales. In all cases, averages were calculated over five replicates ($n = 5$).

components will affect the overall structural properties of nucleoli. Future studies that apply the combination of experimental, computational, and analytical techniques deployed here to more complex systems will enrich our understanding of the relationship between the spatial organization of condensed systems and the network properties.

## Methods

### Cloning

All N130 constructs were subcloned into a pET28b plasmid vector, in frame with an N-terminal 6x His tag, followed by a TEV protease recognition sequence from synthetic double-stranded DNA (Integrated DNA Technologies, Coralville, IA, USA).

### Protein expression and purification

The plasmid constructs were used to transform in *E. coli* strain BL21(DE3) (Millipore Sigma, Burlington, MA, USA), followed by

incubation with shaking at 37 °C. When bacterial cultures reached an optical density at 600 nm of ~0.8, the temperature was reduced to 20 °C and protein expression was induced with the addition of IPTG (GoldBio, St. Louis, MO, USA) to a final concentration of 1 mM, and further incubated with shaking overnight. Cells were harvested by centrifugation and lysed by sonication in buffer A (25 mM Tris, 300 mM NaCl, 5 mM $\beta$-mercaptoethanol, pH 7.5). The soluble fraction was further separated by centrifugation for 30 min at 30,000 × $g$ and loaded on a Ni-NTA column, pre-equilibrated in buffer A. Bound protein was eluted with a gradient of buffer B (25 mM Tris, 300 mM NaCl, 500 mM Imidazole, 5 mM $\beta$-mercaptoethanol, pH 7.5). The fractions containing the protein of interest were identified by SDS-PAGE, pooled and the 6x His affinity tag was removed by proteolytic cleavage, in the presence of TEV protease, while dialyzing against 4 L of 10 mM Tris, 200 mM NaCl, 5 mM $\beta$-mercaptoethanol, pH 7.5. To remove the cleaved affinity tag and any un-cleaved material, the protein was

applied to an orthogonal Ni-NTA column, and the flow-through loaded on a C4 HPLC column, in 0.1% Trifluoroacetic acid, and eluted with a linear gradient of 0.1% Trifluoroacetic acid in acetonitrile. The fractions containing the proteins of interest were identified by SDS-PAGE, pooled and lyophilized. Lyophilized N130 and N130$^{+A2}$ proteins were resuspended in 6 M Guanidine hydrochloride, 25 mM Tris, pH 7.5 and reduced by the addition of 10 mM dithiothreitol. The proteins were refolded by dialysis, using 3 exchanges of 10 mM Tris, 150 mM NaCl, 2 mM mM dithiothreitol, pH 7.5, at 4 °C. The protein concentration during refolding was maintained at or below 100 μM N130 monomer. Protein identities were verified by determining their molecular weight using mass spectrometry in the Center for Proteomics and Metabolomics at St. Jude Children's Research Hospital.

## Fluorescence labeling of proteins

N130 and N130$^{+A2}$ were labeled with Alexa-488 (ThermoFisher, Waltham, MA, USA) at Cys104 by incubating a molar excess of Alexa-488 maleimide with freshly reduced N130 proteins overnight at 4 °C with oscillation. Excess dye was removed by successive rounds of dialysis against 10 mM Tris, pH 7.5, 150 mM NaCl, and 2 mM DTT. Labeled proteins were then unfolded in the presence of 10 mM Tris, pH 7.5, 2 mM DTT, and 6 M GdmHCl and combined with unlabeled protein to a final concentration of 10% labeled protein and refolded by successive rounds of dialysis against 10 mM Tris, pH 7.5, 150 mM NaCl, and 2 mM DTT.

## Fluorescence microscopy measurements

Microscopy plates (Greiner Bio, Kremsmünster, Austria) and slides (Grace BioLabs, Bend, OR, USA) were coated with PlusOne Repel Silane ES (GE Healthcare, Pittsburgh, PA, USA) and Pluronic F-127 (Sigma-Aldrich, St. Louis, MO, USA) and washed with water before the transfer of protein solutions. Fluorescent microscopy experiments were performed using a 3i Marianas system (Intelligent Imaging Innovations Inc., Denver, CO, USA) configured with a Yokogawa CSU-W spinning disk confocal microscope utilizing a ×100/1.45 N.A. Zeiss objective (Zeiss Jena, Germany), a Photometric Prime 95B camera (Teledyne, Seattle, Washington, USA), a ×1.5 additional magnification optovar providing 70 nm pixels, and appropriate excitation and emission band pass filters. With a peak emission wavelength of 550 nm, the Rayleigh resolution for this instrument is 213 nm. Acquisition hardware was controlled using the Slidebook 6 software from 3i. The phase diagram depicted in Fig. 2d was generated by computing the average of the index of dispersion of fluorescent microscopy images of five images per well. The threshold for positive phase separation has been set to 10% of the maximum value. FRAP experiments were performed by bleaching a circular area with a diameter of 1 μm in the center of droplets ($n = 12$) to ~50% of initial fluorescence intensity. The observed fluorescence intensities were then normalized to global photobleaching during data acquisition and fitted as a group to determine recovery times according to[119] Eq. (1):

$$I_t = \frac{I_0 + I_\infty \frac{t}{t_{1/2}}}{1 + \frac{t}{t_{1/2}}} \qquad (1)$$

Here, $I_0$ is the pre-bleach intensity, $I_\infty$ the steady-state, post-bleach intensity, $t$ is the time at which FRAP is measured, and $t_{1/2}$, the time elapsed before half the pre-bleach intensity has been recovered following photobleaching. Uncertainty in the reported half-lives represent the standard error of the fit to the data. FRAP experiments were performed 1 hour after the mixing of components.

## Peptides used in the study

The rpL5 peptide was synthesized in the Macromolecular Synthesis lab at the Hartwell Center, St. Jude Children's Research Hospital. The lyophilized powder was directly reconstituted in buffer, and the pH was adjusted to 7.5 using 1 M Tris base.

## SANS measurements

N130 and N130$^{+A2}$ were buffer exchanged into 10 mM Tris, 150 mM NaCl, 2 mM DTT, in D$_2$O (measured pH, 7.5). Lyophilized rpL5 peptides were resuspended in dialysis buffer. Monodisperse samples of protein only and phase-separated samples with rpL5 were prepared in the dialysis buffer. SANS experiments were performed on the extended $q$-range small-angle neutron scattering (EQ-SANS, BL-6) beam line at the Spallation Neutron Source (SNS) located at Oak Ridge National Laboratory (ORNL). In 30 Hz operation mode, a 4 m sample-to-detector distance with 2.5–6.1 and 9.8–13.4 Å wavelength bands was used[120] covering a combined scattering vector range of 0.006 < $q$ < 0.44 Å$^{-1}$. $q = 4\pi \sin(\theta)/\lambda$, where $2\theta$ is the scattering angle, and $\lambda$ is the neutron wavelength. Samples were loaded into 1 or 2 mm pathlength circular-shaped quartz cuvettes (Hellma USA, Plainville, NY, USA) and sealed. SANS measurements were performed at 25 °C using the EQ-SANS rotating tumbler sample environment to counteract condensate settling. Data reduction followed standard procedures using MantidPlot[121] and drtsans[122]. The measured scattering intensity was corrected for the detector sensitivity and scattering contribution from the solvent and empty cells, and then placed on absolute scale using a calibrated standard[123]. Additional information regarding the data collection and analysis is given in Supplementary Table 1.

## Atomistic MC simulations using the ABSINTH model

For the first step of systematic CG, we performed atomistic MC simulations to obtain a robust description of the conformational ensembles of N130. For this, we employed the ABSINTH implicit solvation model and forcefield paradigm[87]. In this model, all polypeptide atoms and solution are modeled explicitly, and the degrees of freedom are the backbone and sidechain dihedral angles as well as the translational motions of the solution ions, which are spheres. All simulations were performed using version 2.0 of the CAMPARI modeling package (http://campari.sourceforge.net) and the abs_opls_3.2.prm parameter set. The initial structure of N130 was modelled as a pentamer, where the structure of the ODs is based on the coordinates deposited in the protein data bank (PDB ID: 4N8M). The structures of each disordered N-terminal tail (residues 1–18, GSHMEDSMDMDMSPL) and disordered A2 tract (residues 124–133, EDAESEDEDE) were built using CAMPARI. The degrees of freedom internal to the ODs were held fixed during the ABSINTH simulations, reflecting the fact that the domains are well folded and tightly bound to each other. The system is placed in a soft-wall spherical potential with radius 70 Å. We included sodium and chloride ions to mimic the salt concentration of ~20 mM, in addition to neutralizing ions. The simulation temperature was set to 300 K.

For efficient sampling of the conformational ensemble, we first performed simulations based on the so-called excluded volume or EV limit. In this limit, all terms other than the steric repulsions and any dihedral angle terms in the potential functions are switched off. Note that the ABSINTH model uses fixed bond lengths and bond angles. These initializing simulations were performed for 10$^8$ MC steps. We sampled 100 different structures from the EV limit simulations and used each of them as initial structures simulations based on the full potential. Each simulation consists of 10$^8$ MC steps, and the structural information was stored every 5000 steps. Hence, we collected 20,000 snapshots per trajectory, from 100 independent trajectories of atomistic simulations. Next, we performed ABSINTH-based MC simulations for $2.1 \times 10^7$ MC steps with sampling frequency of (5000 steps)$^{-1}$, where the first 10$^6$ MC steps were discarded as equilibration.

## Computations of scattering profiles using CAMPARI

We used the computed the scattering form factor $P(q)$ from snapshots generated using the ABSINTH-based simulations. We excised the

conformations of N130 pentamers, and computed Kratky profiles using the `scattercalc` functionality within the CAMPARI package (http://campari.sourceforge.net). In these calculations, the scattering cross-sections of all atoms are set to be unity. The results we obtain for $P(q)$ are plotted as $\log(P(q))$ versus $\log(q)$ in Supplementary Fig. 1.

## Systematic CG

Our approach to developing CG models involves three steps: First, we choose the resolution for the CG model. Second, we choose the form for the potential functions that describe interactions among pairs of CG sites. And third, we use a Gaussian process Bayesian optimization (GPBO) module to parameterize the model to ensure that the CG model recaptures conformational statistics of the atomistic simulations. In our choice of the model, each residue is modelled as a bead, except for the OD of N130, which by itself forms a large bead with excluded volume. The mass of each bead is determined by the total mass belonging to the specific bead. For example, the OD bead has mass of 47544.8 amu. The position of each bead is set equal to the position of the center of mass in its atomistic representation.

The potential function used for the simulations is decomposed into five different terms in Eq. (2):

$$W = W_{\mathrm{LJ}} + W_{el} + W_b + W_\theta + W_\phi \qquad (2)$$

Each term contains several interaction parameters, which were parameterized using the CAMELOT algorithm[86]. This uses a GPBO module to minimize an objective function, which is defined as the difference between the site-to-site distance distributions generated by atomistic MC simulations based on the ABSINTH model and by CG MD simulations. Within the CAMELOT algorithm, we change the parameters of the potential function for the CG model, perform CG MD simulations to obtain conformational statistics, specifically inter-site distance distributions, quantify the objective function, and iterate until a stationary state is reached for the objective function. In this work, we used conformational statistics derived from the atomistic simulations of structural ensembles of N130 and R-rich peptides as the reference against which the CG model was parameterized. The optimized parameters are given in Supplementary Tables 2–9. The bead types and corresponding amino acid residues are given in Supplementary Fig. 7.

To reduce the computational cost of scanning the parameter space, we grouped several amino acid types into one bead type, following the previous work. For the rpL5 peptide, we grouped different amino acids into three different bead types: charged (K, R, E, D), large (V, F, M, I, L, Y, Q, N, W, H), and small (A, P, G, S, T, C). Each group has its own $\epsilon_i$ value to be determined. For each residue in either the large or small group, we used $\sigma_i$ as twice of the radius of gyration of the specific residue in the atomistic simulations (and consequently $\sigma_i$ is not identical for all residues in the same group; it is residue-dependent). For charged residues, its $\sigma_i$ was left as a free parameter to be determined by the optimization module. Hence, we have four unknown parameters, three $\epsilon_i$ values for the charged, large, and small bead types plus one $\sigma_i$ value for the charged bead type.

## Coarse-grained model for MD simulations

The CG model is summarized in Fig. 1, and the potential function used for the simulations can be decomposed into five different terms given in Eq. (2). Here, as in Eq. (3)

$$W_{LJ,ij} = 4\epsilon_{ij} \left[ \left(\frac{\sigma_{ij}}{r_{ij}}\right)^{12} - \left(\frac{\sigma_{ij}}{r_{ij}}\right)^6 \right], r_{ij} < r_c \qquad (3)$$

is the standard LJ potential with cutoff $r_c = 2.5\sigma$. While we decomposed the two-body interaction parameters into one-body parameters: $\sigma_{ij} = (\sigma_i + \sigma_j)/2$ and $\epsilon_{ij} = \sqrt{\epsilon_i \epsilon_j}$. All energies have units of kcal/mol.

The electrostatic interactions were modeled using a Debye-Hückel potential given by Eq. (4):

$$W_{el,ij} = C\frac{q_i q_j}{\epsilon r_{ij}}\exp\left(-\kappa r_{ij}\right), r_{ij} < r_c \qquad (4)$$

implemented with the lj/cut/coul/debye pair-style in LAMMPS[124] with $\epsilon = 80.0$, $\kappa = 0.1$, and cutoffs $\sigma_1 = 24.1$ and $\sigma_2 = 15.0$ Å. with constant $C$, charges $q$, dielectric constant $\epsilon$, and inverse Debye length $\kappa$. In this work, we used $\epsilon = 80.0$ and $\kappa = 0.1 \text{Å}^{-1}$. The charges were assigned manually; beads for $R$ and $K$ have +1, beads for $D$ and $E$ have −1, and other beads (including the PD bead) have 0.

The bond and angle terms are modeled as harmonic potentials as in Eqs. (5) and (6),

$$W_{b,i} = K_i\left(b_i - b_{0i}\right)^2 \qquad (5)$$

Equation (5) is a quadratic bonded potential implemented as the harmonic bond-style in LAMMPS[124].

$$W_{\theta,i} = K_i\left(\theta_i - \theta_{0i}\right)^2 \qquad (6)$$

Equation (6) shows a quadratic angular term implemented as the harmonic angle-style in LAMMPS[124]. The bond parameters $K_{b,i}$ and $b_{0i}$ were obtained by fitting the normal distribution to the distribution of the distance between two adjacent residues. The angle parameters $K_{a,i}$ and $\theta_{0i}$ were also obtained by fitting the normal distribution to the distribution of the angle between three adjacent residues. For the OD bead, we assigned arbitrarily high values for the energy parameters: $K_{b,i} = 60,000$ kcal/mol-Å² and $K_{a,i} = 60,00,000$ kcal/mol-radian², keeping the bond extremely rigid. The equilibrium length and angle were determined by the PDB structure.

Except for the PD bead, the dihedral term is given by a Fourier series potential shown in Eq. (7),

$$W_{\phi,i} = \sum_{n=1}^{3} K_{ni}(1 - \cos(n\,\phi_{0i} - \phi_{ni})) \qquad (7)$$

This is a Fourier series dihedral term implemented as part of the class2 dihedral style in LAMMPS. with arbitrarily high value of $K_{d,i} = 60,00,000$ kcal/mol-radian² and experimentally determined $\phi_{0i}$. The parameters corresponding to the potentials, derived from CAMELOT, for the different systems considered are in Supplementary Tables 2–9. Lastly, $W_{\phi,i.\mathrm{quad}} = K_i(\phi_i - \phi_{0i})^2$ is used to constrain the five arms of N130 with a very high $K$-values.

## Simulations of the N130 wild type and rpL5 peptides

Given the initial configuration generated using CAMELOT[86], we used the replication command in LAMMPS to generate 108 copies of N130 pentamers 1620 copies of the rpL5 peptide. Following the replication, deform and nve/limit fixes were used to reduce the box sizes for the simulations to 250 nm. The final configuration served as initial conditions for NPT simulations to prepare systems at the correct intrinsic density. NPT simulations were run for $1 \times 10^7$ steps with a timestep of 1 fs. A Nose-Hoover thermostat and barostat were used, with damping constants of 100 and 1000 fs, respectively. The final configurations from these NPT simulations served as the starting configuration for NVT simulations, which were run for $1 \times 10^8$ steps with a timestep of 0.1 fs and a Nose-Hoover thermostat of 10 fs. The velocities were randomized. Trajectory snapshots were output every 50,000 steps, and we only considered the last 1000 frames for the different analyses. Supplementary Fig. 8 shows that the production runs were equilibrated. Five independent NVT replicates were run for each condition, and the standard error of the mean between replicates is used as the measure of uncertainty. The system setup is summarized in Table 1.

**Table 1 | System setup for NVT MD simulations of N130 and rpL5 peptides**

| System | Box size (Å³) | Total number of atoms | Total number of molecules | Number of N130 molecules | Number of rpL5 molecules |
|---|---|---|---|---|---|
| N130 + rpL5 | 338.6 × 338.6 × 338.6 | 50,328 | 1728 | 108 | 1620 |
| A0, mut + rpL5 | 340 × 340 × 340 | 50,328 | 1728 | 108 | 1620 |
| A1, mut + rpL5 | 340 × 340 × 340 | 50,328 | 1728 | 108 | 1620 |
| A2, mut + rpL5 | 340 × 340 × 340 | 50,328 | 1728 | 108 | 1620 |

The naming for the N130 mutants follows Fig. 4.

## Simulations of the N130 mutants and rpL5 peptides

Starting with the initial configuration generated using CAMELOT[86], we used the replication command in LAMMPS as before and the deform and nve/limit fixes to reduce the box sizes to approximately the same dimensions as the simulations of the N130 wild type and rpL5 peptides (Table 1). The charges within each acidic region were neutralized for these simulations. Keeping the charges neutralized, we then mixed the species using the indent fix and ran NVT simulations in LAMMPS for $2 \times 10^8$ steps. We used a timestep of 0.1 fs and a Nose-Hoover thermostat with a damping constant of 10 fs, as before. The velocities were randomized, as before. Trajectory snapshots were output every 50,000 steps in the last $1 \times 10^8$ steps, and only the last 1000 frames were considered for analyses. Supplementary Figs. 9–11 show that the production runs were equilibrated. Five independent NVT replicates were run for each condition, and the standard error of the mean between replicates was used as the measure of uncertainty.

## Simulations of the LJ systems

To understand the network structure of different LJ phases, we performed NVT simulations in LAMMPS, with 10,000 LJ particles. We used the NIST parameters for a pure LJ gas, a pure LJ fluid, and a pure LJ solid to access different pure phases. In reduced units, the densities and temperatures for the different systems are given in Table 2.

With a timestep $dt = 0.005\tau$, where $\tau$ is the dimensionless LJ time unit, an initial 50,000 steps were run to let the systems settle, after which $1 \times 10^7$ steps were run for data production. A Nose-Hoover thermostat was used with a damping constant of $0.5\tau$. A non-bonded cutoff of $2.5\sigma$ was used. The velocities were randomized, as with the N130 + rpL5 simulations. Trajectory snapshots were output every 10,000 steps. The last 500 frames of the production run were used to calculate the RDFs and degree distributions. Supplementary Figs. 12–14 show that the production runs were equilibrated. Three independent replicates were run for each condition, and the standard error of the mean between replicates was used as the measure of uncertainty.

## Calculation of the RDFs $g(r)$

We used VMD[125] to calculate $g(r)$ for the different sets of beads considered in this work. To calculate the $g(r)$ for the N130 + rpL5 system, we used a bin size, $dr$, of 0.5 Å. Note that the $g(r)$ profiles for the N130 mutants were computed from the pair distributions between all the acidic residues in the select acidic region, and all basic residues in the peptide. For the LJ fluid, we used a bin size of 0.025 $dr/\sigma$. As with our analysis of the N130 + rpL5 system, we averaged over all replicates.

## Graph-building methods

First, we identify the regions on different molecules that contribute to the network structure of the fluid we wish to investigate. We focused our analysis on the network formed by acidic residues in a particular acidic region in N130 and the basic residues in rpL5. This defines two sets of residues. Given the two sets of residues/beads, we calculate the inter-set $g(r)$ as explained in the methods. For each $g(r)$ we compute the location of the first minimum. The locations of these minima serve as the cutoff radii for defining the presence of an edge between beads.

**Table 2 | Density and temperature for different LJ phases in reduced units**

| Phase | $\rho^\star$ | $T^\star$ |
|---|---|---|
| Vapor | 0.01 | 1.0 |
| Fluid | 0.8 | 1.0 |
| Solid | 1.5 | 0.758 |

By definition, reduced units imply unitless parameters.

Given two sets of beads, a particular trajectory snapshot, and the computed cutoff radius, we generate a bead adjacency matrix. A basic bead is considered adjacent to an acidic residue if the distance between the selected acidic and basic beads is within the cutoff radius. This calculation is performed for all pairs defined by the two sets of beads that are chosen for the analysis. This generates a bead adjacency matrix where an edge is drawn between beads in the chose set if the inter-bead distances are within the computed cutoff radius. We can either consider the total bead adjacency matrix where every bead in the system is included and where all the bead types not in the initially chosen set are non-adjacent by construction, or we can generate a bead adjacency matrix where only the considered beads are included. We choose the latter.

In more detail, suppose that set-1 has $m$-bead types and set-2 has $n$ bead types. Then, given $\alpha$ N130 molecules, set-1 has $\alpha \cdot m$ beads in total. Similarly, given $\beta$ rpL5 molecules, set-2 has $(\beta \cdot n)$ beads in total. Therefore, the bead adjacency matrix will be an $(\alpha \cdot m + \beta \cdot n) \times (\alpha \cdot m + \beta \cdot n)$ matrix. As an example, suppose we have one N130 molecule, and one rpl5 molecule. This would then give us an $(m + 15n) \times (m + 15n)$ matrix. Furthermore, suppose that the beads are ordered such that the first $m$-rows correspond to the N130 beads, and so the next $n$-rows are the rpL5 beads (since adjacency matrices are symmetric the first $m$-columns would be for N130 and the next $n$-columns would be for rpL5). To go from this bead adjacency matrix to a molecular adjacency matrix we would look at blocks of this bead adjacency matrix. Let the bead adjacency matrix be $\hat{B}$. Using an indexing, $B[0:m, 0:m]$ corresponds to the sub-graph of N130 adjacent beads. In our case, no beads will be adjacent since the graph is intentionally constructed between the acidic residues in N130 and the basic residues in rpL5. Moving on, $B[0:m, m:m + n]$ (or $B[m:m + n, 0:m]$ due to symmetry) corresponds to the sub-graph between the N130 beads and the beads of the first rpL5 molecule. Similarly, if we had more than 1 rpL5 molecule, $B[0:m, m + i*n:m + (i + 1)*n]$ gives us the sub-graph between the N130 beads and the beads of the $i$-th rpL5 molecule. To generate the molecular adjacency matrix, $\hat{A}$ we check if any of the sub-graphs from the bead adjacency matrix are non-empty (or that there are edges in that graph) or that there is a 1 in the $B[0:m, m + i*n:m + (i + 1)*n]$ block. Therefore, $A[0,0] = 0$ in our case by construction, or that molecule-0 and molecule-0 are not adjacent.

Returning to the more general case, we have the bead adjacency matrix that corresponds to a matrix of size $(\alpha \cdot m + \beta \cdot n) \times (\alpha \cdot m + \beta \cdot n)$ where the first $\alpha \cdot m$ rows (or columns) correspond to the beads in N130 molecules, and the last $\beta \cdot n$ rows (or columns) correspond to the beads in rpL5 molecules. We check all the blocks of the matrix, which

correspond to the beads between the different molecules in the system. These correspond to the sub-graphs between the beads of different molecules. For any sub-graph or block that is non-empty, we consider the two corresponding molecules to be adjacent. This gives the molecular adjacency matrix, which should have the shape $(\alpha + \beta) \times (\alpha + \beta)$. This is the final graph that is analyzed. Graph properties are calculated per-snapshot and then averaged over the total set of frames considered.

The molecular adjacency graph is constructed by considering the adjacency between any of the beads from the initially selected set. Since we only care about the acidic and basic beads, by construction this graph avoids self-loops. Furthermore, since we only care about specific blocks of the bead adjacency matrix being non-empty, we only obtain an unweighted graph. If, however, we wanted to obtain the weighted graph, we would simply take the sum of the number of edges in a particular sub-graph or take the sum of that block from the bead adjacency matrix.

On an implementation level, we can skip most of the block reductions by simply asking the following: given the two sets of beads and the cutoff radius, which pairs of beads are adjacent. Then, from the pairs of adjacent beads, we ask which molecules the beads in the pairs come from. Specifically, we identify the molecule-ID of each bead. From this set of molecule-ID pairs, we find the unique pairs. Given these unique pairs of adjacent molecule-IDs, we construct the molecular adjacency matrix since the molecule-IDs directly correspond to the indices in the adjacency matrix. We set those elements of the $(\alpha + \beta) \times (\alpha + \beta)$ matrix to one, and we obtain our molecular adjacency matrix.

### Degree distributions
From a given trajectory snapshot, we generate molecular graphs, $G(V,E)$, where individual molecules are represented as nodes, $V$. To calculate the edges between nodes in a generalizable way, we use the first minima from the RDFs of the given sets of beads. We use signal.find_peaks from the SciPy package[126] to find these minima. We generated RDFs for the acidic/negatively charged beads in N130 from a particular acidic region, and the basic residues in the rpL5 peptide. Given the RDFs, we then use the first minimum as the cutoff radius for the definition of an edge. An edge, $E$, is therefore drawn between two nodes if any of the beads from the considered sets are within the distance corresponding to the first minimum of the g($r$) of interest. We use MDAnalysis[127] to analyze the trajectories to find adjacent molecules. Given these adjacency matrices, we then calculate the degree of each node by calculating the total number of edges for each node, or by calculating the sum of each row (or column) of the adjacency matrix. The python package NumPy[128] is used to generate these degrees. From the degrees, we calculate the degree distribution. The degree distributions from the last $10^3$ frames are averaged over, and the average over the five simulation replicates are reported here.

### Mean square displacements
The MSD was calculated by averaging the displacements in particle positions over all windows of length $m$ and over all particles $N$ as shown in Eq. (8):

$$MSD(m) = \frac{1}{N} \sum_{i=1}^{N} \frac{1}{N-m} \sum_{k=0}^{N-m-1} \left( \vec{r}_i(k+m) - \vec{r}_i(k) \right)^2 \qquad (8)$$

for $m = 1, \ldots, N - 1$. For the acidic regions and rpL5, the MSD was calculated with respect to all acidic and basic residues, respectively. For all MSDs, we fit the first 30 ps and everything past 400 ps using single exponents to describe the two different regimes. For the MSD with respect to the PD, the region from ~80–120 ps was found to fit best to a simple diffusion model with $R^2 = 0.9987$. The crossover $t_D$ between the super- and sub-diffusive regimes is the median value within the

diffusive region of the MSD with respect to the PD. In all cases, the MSD was averaged over five replicates. To calculate the histograms of the exponents of the individual molecules, we modified Eq. (2) to calculate the MSD with respect to each PD rather than averaging over all PDs. We then fit each MSD as before.

### Plotting
To generate the plots, we use Matplotlib[129] along with Seaborn[130]. Adobe Illustrator® is used to generate the final figures shown.

### Reporting summary
Further information on research design is available in the Nature Portfolio Reporting Summary linked to this article.

## Data availability
Source data are provided as a Source Data file with this manuscript and via the GitHub repository of the Pappu lab https://github.com/Pappulab/n130-liquid-structure/. Input files for the simulations and coordinate files of the final outputs are available via Zenodo at https://zenodo.org/doi/10.5281/zenodo.10823199[131]. PDB 4N8M is available from the Protein Data Bank. Source data are provided with this paper.

## Code availability
All custom-made code for the analyses can be found on the GitHub repository of the Pappu lab at https://github.com/Pappulab/n130-liquid-structure/. Python (v3.9), VMD (v1.9.3), and MATLAB (r2021b) were used for data analysis. All CAMPARI simulations were performed using version 2.0 available at http://campari.sourceforge.net. All CAMELOT simulations were performed using version 0.1.2. All MD simulations were performed in LAMMPS (16 Dec. 2013).

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

## Acknowledgements

This work was supported by the St. Jude Children's Research Hospital Research Collaborative on the Biology and Biophysics of RNP granules (to R.V.P. and R.W.K.), the US National Science Foundation (MCB-2227268 to R.V.P.), the US National Institutes of Health (NIGMS R01 GM115634 and R35 GM131891 to R.W.K., and NCI P30 CA021765 to St. Jude Children's Research Hospital), ALSAC (supporting studies at St. Jude Children's Research Hospital), and National Research Foundation (NRF) grants of Korea (2021R1C1C1010943 and 2022R1A4A1033471 to J.-M.C.). A portion of this research, conducted at the Oak Ridge National Laboratory (ORNL) Spallation Neutron Source, was sponsored by the Scientific User Facilities Division, Office of Basic Energy Sciences, U.S. Department of Energy. S.R. Cohen acknowledges financial support via T32 EB028092 from the US National Institutes of Health. We thank Jared M. Lalmansingh for technical assistance with CAMPARI. Fluorescence microscopy images were acquired at the St. Jude Cell & Tissue Imaging Center at St. Jude Children's Research Hospital (supported by P30 CA021765); we thank V. Frohlich, J. Peters, A. Taylor, A. Pitre, and G. Campbell for technical assistance.

## Author contributions

R.V.P., J.-M.C. and R.W.K. came up with the project idea. D.M.M. and A.H.P. prepared samples for measurements. D.M.M., A.H.P., W.C.L., G.N., C.B.S., performed SANS measurements and analyzed SANS data. D.M.M. and A.H.P. designed and characterized the phase behaviors of mutants. J-M.C prototyped the CAMELOT-based CG, and the original simulations using LAMMPS. F.D. designed and performed all the LAMMPS simulations reported in this work. F.D., S.R.C. and R.V.P. designed and iterated on the structure of the analysis with inputs from J.-M.C. F.D. and S.R.C. deployed the entirety of analyses, including the SANS data, and integrated the findings with experimental work. F.D., S.R.C. and A.H.P. made the figures. F.D., S.R.C. and R.V.P. wrote the manuscript. All authors contributed to editing of the manuscript.

## Competing interests

R.V.P. is a member of the scientific advisory board and shareholder of Dewpoint Therapeutics Inc. D.M.M. is an employee and shareholder of Dewpoint Therapeutics. The work reported here was not influenced by these affiliations. The remaining authors have no competing interests to declare.
