## [Peer Review File · Nature Communications]

Biomolecular condensates form spatially inhomogeneous network fluidsREVIEWER COMMENTS

Reviewer #1 (Remarks to the Author):

In this manuscript, Dar and coworkers combine SANS and FRAP experiments with coarse-grained computations to study the structural and dynamic organization of model biomolecular condensates that mimic nucleolar granular components. It is shown that the condensates formed by nucleophosmin (NPM1) oligomerization domain (OD), which comprises of the 130 N-terminal residues (N130) of NPM1, and the disordered arginine-rich peptide rpL5 display features of network fluids, similar to known associative polymer systems such as hairy colloids. More specifically, correlation peaks on the SANS curves are reasonably well reproduced by the computations (see more detailed critical comment below), showing that the condensates exhibit short-to-intermediate range order on the length scale of the OD-domain itself, but no long-range order. This interesting behavior is explained by the computations, which identified a hierarchy of interactions, especially between three disordered acidic regions of N130 (A0, A1, A2) and the rpL5 peptide. Further detailed graph-theoretic analyses show that there are two types of sub-graphs behind the structure of N130/rpL5 condensates, a “gas-like” and a “liquid-like” one, which also have distinct dynamic fingerprints. The findings are very relevant and the underlying elegant concepts, as brought to light in this careful study, can be generalizable also to other related systems.

A particularly strong point of the manuscript is also that the authors use their sequence-based coarse-grain model to make predictions that they then tested (and verified) experimentally: They replaced the A0 region of N130, which was found to be involved in the heterotypic interactions, with the residues of the A2 region (in reverse sequence order). The titration experiments then show that this so-called “N130+A2” variant, which has even more acidic residues than the wild-type, indeed forms condensates at a lower rpL5 threshold concentration (Figs. 1 and 2).

We add a number of detailed point below, in the hope that these can help the authors to further clarify and improve their manuscript. These are listed in the order of appearance in the manuscript.

1) p. 8. The $g(r)$ of Lennard-Jones fluids is well-know and established since many decades, and uninteresting from a scientific viewpoint in the context of the current manuscript. We understand that the authors included this here as a reference, but in our opinion it is neither constructive nor necessary to elaborate on this on that much detail in the figure (Fig. 3a) and the text. Thus, the paragraph starting “To calibrate our expectations, ...” is suggested to be moved to SI, together with Fig. 3a.

2) p. 8, bottom: It is stated that the $g(r)$ is computed “between pairs of ODs of different N130 molecules (Fig. 3b).” Is 1 coarse-grained bead per N130 molecule used, or 1 bead per N130 pentamer? From the Methods we understood that it is the latter, so we guess this would need to be corrected here (not “between N130 molecules” but “between N130 pentamers”).

3) p. 9, top: Based on the similarity of the coordination number of close to 4 to that of liquid water

(coordination number of ca. 4.5), the authors suggest “that N130 forms a network fluid that resembles the open structure of liquid water.” However, we are not sure whether this similarity of the coordination numbers is really enough to make this statement (it could just be coincidence, or even if it is not, it does not necessarily mean that there is any similarity between liquid water and the N130 OD network fluid...). So the question is: Is there additional evidence to support this analogy? If not, this should be more carefully written in our opinion, because it might otherwise open the door for undesired misunderstandings.

4) p. 9, top: The peaks in the computational $g(r)$ are at 52 Å, 100 Å, and 145 Å, and are stated to be “in reasonable agreement with the molecular spacing indicated in our SANS data.”. However, the SANS peaks are at 55 Å, 77 Å, and 119 Å, and thus at least the latter two are quite substantially off (23 – 25 Å). We think that a) these clear deviations should be more critically discussed, including the (possible) underlying reasons, and b) that the interpretations of the results should take these deviations into account as well.

5) p. 14, bottom: The authors conclude that “The sub-diffusive dynamics derived from simulations are consistent with the apparent plateau in the FRAP data for the wild type (Fig. 2d). First, Fig. 2d does not show and FRAP data (we guess the authors want to refer to another Figure here). More importantly, it is unclear to us how *exactly* the sub-diffusive dynamics matches the FRAP data? In general, it is expected that sub-diffusive regime will be found on time scales that are shorter than the time needed for the molecules to diffuse a mean path length (corresponding to a mean squared displacement) that is on the order of the size of the aggregate, because the molecules experience confinement inside the aggregates (but not when they leave into the other, “gas-like” phase). Can this be confirmed by the actual data?

6) p. 18, middle: N130 was modeled as a pentamer. Why? Can the authors motivate their choice? Why were smaller oligomers (or even monomers) excluded/neglected?

7) p. 18, bottom: In the initial simulations in the excluded volume (EV) limit, all dihedral angle terms in the potential energy functions were switched off. But doesn't that mean that cis/trans configurations of the peptide bonds get screwed up, in the sense that after the EV run one has 50% cis and 50% trans peptide bonds? As the authors know, most peptide bonds should be in the trans configuration, and “repairing” an out-of-equilibrium starting distribution (50/50 cis/trans, see above) could take very many (too many?) subsequent MC steps?

8) p. 20/21: We are a bit puzzled by the “Lennard Jones fluids” and “Simulating of [sic!] the N130 wild type and rpl5 peptide variants” parts. Previously, the Monte Carlo simulations are described, but now the authors turn to molecular dynamics (MD) simulations as far as we understand. This transition should be more clearly explained in the text, also including the timesteps used for integration. Furthermore, “3 x 3 x 4 = 108 copies” is probably a mistake?

Possible Typos:

a) p. 4, Fig. 1 caption: Is the statement “... peaks at scattering vectors corresponding to 55 Å (left arrow), 77 Å (middle arrow), and 119 Å (right arrow).” correct, or are these in reverse order (right, middle, left)?

b) p. 7: "... we replaced A0 with the residues from, in reverse order, ..."
 "... we replaced A0 with the residues from A2, in reverse order, ..."

Reviewer #2 (Remarks to the Author):

The paper from Dar F. and coworkers presents a study on model condensates that mimic nucleolar granular components (GCs). The Authors use a combination of experimental approaches and molecular dynamics simulations. The paper is built on previous observations of the phase behavior of the N130-rpL5 system, eventually discussing the coexistence of liquid- and gas-like phases affecting the internal dynamics. The paper is written in a concise way.

Regarding the interpretation of the data, we have several points, especially in connection with the experimental part. Please see the detailed comments below.

1) The data presented in Figure 1 reproduces the effect of complexation between the acidic regions of N130 and the rpL5. A similar set of data is also presented in the work from Mitrea et al, Elife 2016 (<https://elifesciences.org/articles/13571>), which is within the reference list and involves some of the Authors of the present study. Specifically, in Figure 1b and 3b of the mentioned paper, both the phase diagram and the neutron data can be found. From a closer look at the data for the phase diagram, we acknowledge the fact that the concentration range was slightly extended for both N130 and the rpL5, but overall, we fail to see the degree of novelty represented by Figure 1 in the present paper. Could the Authors elaborate on the novelty of the results in Figure 1?

2) The phase diagram (Figure 1 d) depicts the sample at 400 μM rpL5 and 75 μM N130 as not being phase separated, however for the same concentration of N130, at both 350 and 450 μM (i.e. above and below the 400 μM conc of rpL5), condensed phases are observed. This is perhaps unexpected, could the Authors further comment on this? Can this be due to the stability of droplets?

3) According to the caption for Figure 1e the distances in the SANS spectra correspond to 55 angstrom, 77 angstrom and 119 angstrom going from left to right, this should instead be from right to left, as also previously reported by Mitrea et al, Elife 2016 (<https://elifesciences.org/articles/13571>). Moreover, in the study of Mitrea et al of a similar system, the peak position is determined by Gaussian fitting of the spectra. It is not clear from the methods section how the Authors obtain the peak positions they described. Especially for the peak position at the lowest q value (corresponding to 118 angstrom) it is not obvious (at least from the shown spectra) that there is a well-defined maximum.

4) On page 5 the authors state 'The importance of complexation as a drive of internal organization is made clear by the lack of peaks in the structure factor'. However, the procedure for the structure factor determination from the data is not reported. In the absence of inter-protein interactions, the structure

factor is equal to 1 and the scattering shown for N130 in figure 1e is caused purely by the form factor scattering. It would be relatively straightforward to model the scattering data shown for pure N130, and confirm that the observed scattering is consistent with the expected size/shape of the protein and thus there is an absence of structure factor. Without a more quantitative approach towards the data, their interpretation may result weak. Authors should provide in the methods section a clearer overview of the procedure used for the data analysis and clarify the importance of the SANS data in their work.

5) Regarding the SANS data in Figure 2, the Authors state that in the A2 mutant a broad peak appears at intermediate q . However, the interpretation of this is a bit vague. It would be more cautious not to draw many conclusions from these SANS data as the measured intensity is a product of the form and structure factor of all components in the solution. The new variant (A2+) is not studied in detail, it is likely that the variant has a different form-factor compared to the WT. To address this aspect, SAXS/SANS experiments should be performed on both the WT and A2 mutant in order to obtain the experimental form-factor.

6) A quantitative evaluation of the peak position is also extremely relevant in relation to the data in Figure 3b. Indeed, the peak assignment from the simulations (53 Å, 95 Å and 140 Å) is considered in agreement with the SANS data (see caption of Figure 3b). However, between simulations and experiments there is a significant difference at least for two peaks (SANS peak estimation: 55, 77, 119 Å). Moreover, in the main text (page 9, first line) values of 52 Å, 100 Å and 145 Å are reported for what seems to be the analysis of simulations data in Figure 3b. If this is the case, could the Authors explain the mismatch between the peak positions from simulations reported in the main text and the ones reported in the caption?

7) Regarding the FRAP data, it would be beneficial to elaborate on how the recovery time is obtained. From the experimental data, the curves for WT and the A2 variant largely overlap if one considers the error bars. The reported data in page 7 only show errors in the range of 4-8%. Could the Authors comment on the significance of the calculated values compared to experimental data shown?

8) Page 7 second paragraph, here the mutant N130a+2 is introduced. The text states “we replaced A0 with the residues from, in reverse order, to increase linear charge density”. Shouldn't it be with residues from A2? In the method section it is also not stated precisely what this new variant of N130 contains.

9) For the SANS data of the variants in Figure 5 it would be useful to compare the effects the mutations have on the form-factor of the rpl5 variants (i.e., adding the experiments in the absence of N130). This would support the statement of the Authors that the changes in low- q are due to changes in the shape of the protein. This could be achieved by measuring SAXS or SANS of just the variants at different concentrations to obtain the experimental form factor. It would also be useful to have a proper analysis of the peak positions. The second peak at around 0.13 angstrom seems to initially move to slightly higher- q with increasing L but then for 20L it shifts to lower q . This combined with the drastic change at low q , observed for 20L, might suggest that the structure-factor is the measured scattering pattern, which could potentially cause the shift of the peak position at high- q combined with the broadening at low- q .

10) In Figure 5, we can observe quite significant differences in the signal-to-noise ratio (especially at high

q) of the SANS data (i.e., please compare the error on 10 L with 20L in figure 5a). Because it depicts a comparison with the SANS data in figure 1, one assumes that the same concentrations were used (of both proteins). However, this is not clear from either the caption, text or methods. If the same concentrations were used perhaps there was a different exposure time between the samples? Overall, it is hard to make a comparison between the data in this figure. Showing the data with an offset in the y-axis between the samples may clarify the statements about the data made.

11) Regarding the data in Figure 8. The Authors state that “The sub-diffusive dynamics derived from simulations are consistent with the apparent plateau in the FRAP data for the wild type (Fig. 2d) at long timescales”. Could the Authors elaborate more on this correlation between simulations and experiments? This is also related to our comment 7.

In summary, in the present form, the paper lacks the adequate amount of information to evaluate the robustness of the experimental work performed. Moreover, a set of extra experiments would be greatly beneficial for supporting some of the claims of the work and further appreciating potential consistency with simulations. In general, a clearer discussion on the connection between experiments and simulations should be added. Together with the specific concerns highlighted above, it is also fundamental for the Authors to elaborate more on the novelty of their results, both at the level of the introduction and in terms of research questions, and in the discussion part. We hope that the present report may be functional for the improvement of the manuscript.

Reviewer #3 (Remarks to the Author):

The paper by Dar et al provides an experimental and computational study of the multiscale structure of biomolecular condensates. Using in vitro experiments of N130+rpL5 condensates (that mimic nucleolar granular components) and computer simulations the authors show that these biomolecular condensates are network fluids with spatially inhomogeneous structure across different scales. Overall, this is a very interesting and well-written paper that contributes new insights into the emerging picture that biomolecular condensates are not spatially homogeneous liquids but rather viscoelastic materials with network-like organisation. I recommend publication in Nature Communications. I have a few minor comments, which I hope the authors will find useful in revising the manuscript:

- 1) In Fig 7, the authors show that N130+rpL5 condensates have bimodal degree distributions and hence show coexistence of gas- and liquid-like organization. How does one interpret the freely diffusing gas phase? Would this be similar to the dilute phase (which is “equilibrated” across the condensate interface)?
- 2) What determines the timescale t_D separating the super-diffusive and sub-diffusive regimes in the network fluids?
- 3) In the conclusions, it would be interesting if the authors could expand the discussion on the nucleolus as a multiphase condensate. How do the authors expect the network structure to vary for the various phases in the nucleolus?

Responses to Reviewers

We thank the three reviewers for their assessments of our manuscript. We have made extensive revisions to the manuscript to respond to the comments, concerns, and critiques. For ease of tracking, the revised text is marked in red in the new manuscript. In the current document, the comments made by reviewers are shown in *italics*, our responses are in normal text, and excerpts from the revisions are reproduced in red font.

Responses to Comments of Reviewer 1

Summary comments: *In this manuscript, Dar and coworkers combine SANS and FRAP experiments with coarse-grained computations to study the structural and dynamic organization of model biomolecular condensates that mimic nucleolar granular components. It is shown that the condensates formed by nucleophosmin (NPM1) oligomerization domain (OD), which comprises of the 130 N-terminal residues (N130) of NPM1, and the disordered arginine-rich peptide rpL5 display features of network fluids, similar to known associative polymer systems such as hairy colloids. More specifically, correlation peaks on the SANS curves are reasonably well reproduced by the computations (see more detailed critical comment below), showing that the condensates exhibit short-to-intermediate range order on the length scale of the OD-domain itself, but no long-range order. This interesting behavior is explained by the computations, which identified a hierarchy of interactions, especially between three disordered acidic regions of N130 (A0, A1, A2) and the rpL5 peptide. Further detailed graph-theoretic analyses show that there are two types of sub-graphs behind the structure of N130/rpL5 condensates, a “gas-like” and a “liquid-like” one, which also have distinct dynamic fingerprints. The findings are very relevant and the underlying elegant concepts, as brought to light in this careful study, can be generalizable also to other related systems. A particularly strong point of the manuscript is also that the authors use their sequence-based coarse-grain model to make predictions that they then tested (and verified) experimentally: They replaced the A0 region of N130, which was found to be involved in the heterotypic interactions, with the residues of the A2 region (in reverse sequence order). The titration experiments then show that this so-called “N130+A2” variant, which has even more acidic residues than the wild-type, indeed forms condensates at a lower rpL5 threshold concentration (Figs. 1 and 2). We add a number of detailed point below, in the hope that these can help the authors to further clarify and improve their manuscript. These are listed in the order of appearance in the manuscript.*

Response to summary comments: We thank the reviewer for their careful reading of our manuscript, for their insightful comments, and critical evaluation. We have made all the changes requested by the reviewer, and our revisions were greatly enabled by the depth, precision, and insights that characterized all the comments made by the reviewer.

Comment 1: *p. 8. The $g(r)$ of Lennard-Jones fluids is well-know and established since many decades, and uninteresting from a scientific viewpoint in the context of the current manuscript. We understand that the authors included this here as a reference, but in our opinion it is neither constructive nor necessary to elaborate on this on that much detail in the figure (Fig. 3a) and the text. Thus, the paragraph starting “To calibrate our expectations, ...” is suggested to be moved to SI, together with Fig. 3a.*

Response to comment 1: Please note that there are numerous occurrences of the term “simple liquid” to describe the liquid state of condensates. The criteria used to make these adjudications

viz., roundedness of a condensate, fast recovery of fluorescence after photobleaching, uniformity of density as measured using diffraction-limited methods, and sensitivity to solutes are all *ad hoc* and do not qualify as criteria that would come under the rubric of any formal, thermodynamic description of liquids. Furthermore, as noted in references 9 and 10, and so many other papers that have followed, condensates are described as being “simple liquids” as opposed to “complex fluids”. This is plain wrong, and it is why we thought it essential to take up one figure panel to introduce the cognoscenti and those being initiated to the field to the distinction between formal definitions versus colloquial usages of “simple liquids”. However, we fully understand the reviewer’s position. We have deleted Fig. 3a and moved it to the supplementary material. We have shortened the discussion of the Lennard-Jones (LJ) system, at least as far as the analysis of radial distribution functions is concerned.

Comment 2: *p. 8, bottom: It is stated that the $g(r)$ is computed “between pairs of ODs of different N130 molecules (Fig. 3b).” Is 1 coarse-grained bead per N130 molecule used, or 1 bead per N130 pentamer? From the Methods we understood that it is the latter, so we guess this would need to be corrected here (not “between N130 molecules” but “between N130 pentamers”).*

Response to comment 2: This is an excellent catch, and we regret the snafu on our part. We have changed all occurrences of OD to PD for *pentamerized domain*.

Comment 3: *p. 9, top: Based on the similarity of the coordination number of close to 4 to that of liquid water (coordination number of ca. 4.5), the authors suggest “that N130 forms a network fluid that resembles the open structure of liquid water.” However, we are not sure whether this similarity of the coordination numbers is really enough to make this statement (it could just be coincidence, or even if it is not, it does not necessarily mean that there is any similarity between liquid water and the N130 OD network fluid...). So the question is: Is there additional evidence to support this analogy? If not, this should be more carefully written in our opinion, because it might otherwise open the door for undesired misunderstandings.*

Response to comment 3: Associative fluids belong to the same class at least via the principle of corresponding states even if they are not part of the same universality class. However, the reviewer’s point is well-taken, and we do not wish to lean hard on the similarity, which is not a coincidence, but is nevertheless a distraction. Therefore, we have deleted all mentions, save for one occurrence on line 293.

Comment 4: *p. 9, top: The peaks in the computational $g(r)$ are at 52 Å, 100 Å, and 145 Å, and are stated to be “in reasonable agreement with the molecular spacing indicated in our SANS data.”. However, the SANS peaks are at 55 Å, 77 Å, and 119 Å, and thus at least the latter two are quite substantially off (23 – 25 Å). We think that a) these clear deviations should be more critically discussed, including the (possible) underlying reasons, and b) that the interpretations of the results should take these deviations into account as well.*

Response to comment 4: In the revised manuscript, we have clarified the quality of the agreement / disagreement between inferences from the SANS data and direct assessments from the simulations. The relevant text appears on line 296 and is reproduced below:

The peaks in $g_{\text{PD-PD}}(r)$ occur at 53 Å, 95 Å, and 144 Å. The second and third peaks correspond to ordering beyond the molecular length scale. The ratios of the computed peaks to those estimated based on SANS measurements are 0.96 and 1.25 for the first and second, peaks, respectively. Note that the estimates of higher-order peaks from SANS data are less reliable given lower signal-to-

noise as quantified using analysis of the derivatives (Supplementary Fig. S2). Further, the parameters of the CG model, especially the parameters for Van der Waals interactions, which are governed by the inter-residue and inter-domain distances, will depend on the screening length and ion-mediated correlations in atomistic simulations. The ABSINTH model includes explicit representations of solution ions, and these simulations were performed at low ionic strength, with the salt concentration set at 20 mM given the explicit representations of ions and large droplet sizes. The inclusion of explicit representations of ions leads to exponential increases in simulation time because of the way electrostatic interactions are handled in the ABSINTH model⁸⁷. In the SANS measurements, the salt concentrations were 150 mM. Therefore, given the parameterization of the CG model using atomistic simulations, the differences in peak positions that correspond to intermediate and longer-range ordering are due to differences in effective Debye lengths between the simulations and SANS measurements. Because the Van der Waals parameters are learned from atomistic simulations, one cannot achieve perfect congruence by simply changing Debye lengths in the coarse-grained simulations. Instead, we need salt concentration dependent parameters within the CG model. This requires a model for how the salt-dependent interactions change at different length scales. The remainder of the discussion focuses on insights we can glean from the CG simulations. In doing so, we presume semi-quantitative rather than fully quantitative congruence with SANS experiments.

Comment 5: *p. 14, bottom: The authors conclude that “The sub-diffusive dynamics derived from simulations are consistent with the apparent plateau in the FRAP data for the wild type (Fig. 2d). First, Fig. 2d does not show FRAP data (we guess the authors want to refer to another Figure here). More importantly, it is unclear to us how *exactly* the sub-diffusive dynamics matches the FRAP data? In general, it is expected that sub-diffusive regime will be found on time scales that are shorter than the time needed for the molecules to diffuse a mean path length (corresponding to a mean squared displacement) that is on the order of the size of the aggregate, because the molecules experience confinement inside the aggregates (but not when they leave into the other, “gas-like” phase). Can this be confirmed by the actual data?*

Response to comment 5: The reviewer is correct. In hindsight, we made the same error that is made by many in the condensate field. FRAP data, unlike single particle tracking data, do not really tell us much about molecular transport. Instead, it is becoming clear that these data are readouts of convolutions of molecular transport and the lifetimes of physical crosslinks, with an outsized contribution being made by the latter. Therefore, we have deleted any mention of FRAP in the context of our discussion.

Comment 6: *p. 18, middle: N130 was modeled as a pentamer. Why? Can the authors motivate their choice? Why were smaller oligomers (or even monomers) excluded/neglected?*

Response to comment 6: Mitrea et al. showed that NPM1 and N130 form constitutive pentamers at the salt concentrations used in all our measurements. The data are available in Ref. 78.

Comment 7: *p. 18, bottom: In the initial simulations in the excluded volume (EV) limit, all dihedral angle terms in the potential energy functions were switched off. But doesn't that mean that cis/trans configurations of the peptide bonds get screwed up, in the sense that after the EV run one has 50% cis and 50% trans peptide bonds? As the authors know, most peptide bonds should be in the trans configuration, and “repairing” an out-of-equilibrium starting distribution (50/50 cis/trans, see above) could take very many (too many?) subsequent MC steps?*

Response to comment 7: It appears that the relevant text was misread or misunderstood. Please find the following text, which appears on line 603 of the revised manuscript: **In this limit, all terms other than the steric repulsions and any dihedral angle terms in the potential functions are switched off.** We do not turn off the dihedral angle terms, so there are no issues with *cis* to *trans* isomerization.

Comment 8: *p. 20/21: We are a bit puzzled by the “Lennard Jones fluids” and “Simulating of [sic!] the N130 wild type and rpL5 peptide variants” parts. Previously, the Monte Carlo simulations are described, but now the authors turn to molecular dynamics (MD) simulations as far as we understand. This transition should be more clearly explained in the text, also including the timesteps used for integration. Furthermore, “3 x 3 x 4 = 108 copies” is probably a mistake?*

Response to comment 8: We have added the requested clarifications and corrected the numerical error. There indeed are 108 copies of N130 pentamers and 1620 copies of rpL5 molecules. We have specified these numbers in the main text and in the methods section.

Responses to comments of Reviewer 2

Summary comments: *The paper from Dar F. and coworkers presents a study on model condensates that mimic nucleolar granular components (GCs). The Authors use a combination of experimental approaches and molecular dynamics simulations. The paper is built on previous observations of the phase behavior of the N130-rpL5 system, eventually discussing the coexistence of liquid- and gas-like phases affecting the internal dynamics. The paper is written in a concise way. Regarding the interpretation of the data, we have several points, especially in connection with the experimental part. Please see the detailed comments below.*

Response to summary comments: We thank the reviewer(s) for their inputs. Below, we provide point-by-point responses and an inventory of the revisions we have made in response to the reviewer’s comments.

Comment 1: *The data presented in Figure 1 reproduces the effect of complexation between the acidic regions of N130 and the rpL5. A similar set of data is also presented in the work from Mitrea et al, Elife 2016 (<https://elifesciences.org/articles/13571>), which is within the reference list and involves some of the Authors of the present study. Specifically, in Figure 1b and 3b of the mentioned paper, both the phase diagram and the neutron data can be found. From a closer look at the data for the phase diagram, we acknowledge the fact that the concentration range was slightly extended for both N130 and the rpL5, but overall, we fail to see the degree of novelty represented by Figure 1 in the present paper. Could the Authors elaborate on the novelty of the results in Figure 1?*

Response to comment 1: The central objective of our work was to provide a molecular-scale description of intra-condensate structure. This we achieved by adapting methods from liquid-state theory, which have rested on combining inferences from scattering measurements and making sense of these using simulations and / or theory. We do not expect readers to be fluent with all the details of prior publications, and the content of Figure 1 is included to provide context and continuity. We did not claim novelty and we did cite the work of Mitrea et al. So, we do not see any difficulties with the inclusion of Figure 1. The following text in the introduction, starting on line 101 of the revision, provides motivation for our work:

Here, we revisit the SANS data collected by Mitrea et al.,⁷¹ updating these with new measurements and combining these with simulations to answer the following question: how might

descriptors from theories of simple and complex fluids be adapted for describing condensates? To answer this question, we adapt approaches that integrate scattering data with computer simulations^{78, 79, 80, 81, 82, 83, 84, 85}. We combine traditional approaches based on pair distribution functions with graph-theoretic methods to arrive at descriptions of network structures of condensates formed by N130 and rpL5. The simulations we use are based on bespoke, sequence-specific coarse-grained models. The latter were developed using a machine-learning approach that is bootstrapped against atomistic simulations⁸⁶.

Comment 2: The phase diagram (Figure 1 d) depicts the sample at 400 μM rpL5 and 75 μM N130 as not being phase separated, however for the same concentration of N130, at both 350 and 450 μM (i.e. above and below the 400 μM conc of rpL5), condensed phases are observed. This is perhaps unexpected, could the Authors further comment on this? Can this be due to the stability of droplets?

Response to comment 2: Microscopy-based assessments of phase boundaries tend to have issues at the phase boundary due to the noise floor. We have included the raw data that we collected. We also collected additional data and reanalyzed the data to include more replicates. The revised phase boundary is now included in Figure 1d, and the revised figure and caption are pasted below.

Fig. 1: Complexation between acidic regions within N130 and R-motifs of rpL5 is required for condensation. (a) Schematic representation of N130 including the different acidic regions. The amino acid sequence of N130 is also shown. The three acidic regions A0, A1, and A2 span residues 4-18, 35-44, and 120-133, respectively. Non-native N-terminal residues remaining after protease cleavage are underlined. On the right, we show the overall structure of the hairy colloid generated by superposition of 50 distinct conformations from atomistic simulations. These simulations use the ABSINTH implicit solvent model⁸⁷ and explicit representations of solution ions⁸⁸ (which are not shown in the figure). The pentamerized OD (PDB ID 4N8M), in gray, was modeled as a rigid molecule in the atomistic simulations. (b) The amino acid sequence of rpL5. The panel on the right shows a superposition of 50 different conformations extracted from ABSINTH-based simulations. (c) Confocal microscopy images of phase separation of 100 μM N130 upon titrating the concentration of rpL5 in buffer and 150 mM NaCl. N130 is labeled with green.

AlexaFluor488. **(d)** Two-component phase boundary for N130 + rpL5, showing the result of concentration titrations. **(e)** SANS curve showing the intensity $I(q)$ plotted against q , the scattering vector, for condensates formed by a solution of N130 (200 μ M): rpL5 at 1:3 stoichiometry. Multi-peak analysis, as described by Mitrea et al.,⁷¹ leads to the identification of two major peaks corresponding to 55 Å (right arrow) and 77 Å (left arrow) that are annotated on the figure. The SANS curve for N130 pentamers, in the absence of rpL5, is shown for comparison. In the interest of clarity, this curve is shifted upwards vis-à-vis the curve for the N130 + rpL5 system. We computed the scattering curve for individual N130 pentamers. These computed profiles show qualitative resemblance to the SANS curve shown here for N130 pentamers (see Supplementary Fig. S1).

Comment 3: *According to the caption for Figure 1e the distances in the SANS spectra correspond to 55 angstrom, 77 angstrom and 119 angstrom going from left to right, this should instead be from right to left, as also previously reported by Mitrea et al, Elife 2016 (<https://elifesciences.org/articles/13571>). Moreover, in the study of Mitrea et al of a similar system, the peak position is determined by Gaussian fitting of the spectra. It is not clear from the methods section how the Authors obtain the peak positions they described. Especially for the peak position at the lowest q value (corresponding to 118 angstrom) it is not obvious (at least from the shown spectra) that there is a well-defined maximum.*

Response to comment 3: We regret the error in identifying the length scales on the scattering profiles. We have fixed this error. We have included an analysis of the derivatives of the $I(q)$ versus q data. Please see Supplementary Fig. S2. As the reviewer notes, the assignment of additional peaks becomes confounded by low signal-to-noise in the data. The issue is already evident even with the assignment of the second peak. Therefore, we have decided against assigning a third peak in Fig. 1e.

Comment 4: *On page 5 the authors state ‘The importance of complexation as a drive of internal organization is made clear by the lack of peaks in the structure factor’. However, the procedure for the structure factor determination from the data is not reported. In the absence of inter-protein interactions, the structure factor is equal to 1 and the scattering shown for N130 in figure 1e is caused purely by the form factor scattering. It would be relatively straightforward to model the scattering data shown for pure N130, and confirm that the observed scattering is consistent with the expected size/shape of the protein and thus there is an absence of structure factor. Without a more quantitative approach towards the data, their interpretation may result weak. Authors should provide in the methods section a clearer overview of the procedure used for the data analysis and clarify the importance of the SANS data in their work.*

Response to comment 4: We have included a new figure in the supplementary material – please Supplementary Fig. S1. Here, we show the scattering curve computed using our own CAMPARI software (<http://campari.sourceforge.net>). The structures for N130 pentamers were taken from the ABSINTH-based atomistic simulations. In the calculations, we assume identical scattering cross-sections for all atoms. The computed scattering curve is intended to assess whether the shape of the SANS curve for N130 pentamers can be reliably attributed to this molecule and to the pentameric form. The curve, reproduced below, shows good agreement with the profile shown in red in Fig. 1e. Details of how the scattering curves were computed are included in the revised methods section.

Supplementary Fig. S1: Scattering probability using CAMPARI for N130 pentamers. Snapshots from ABSINTH-based atomistic simulations of N130 in the presence of rpL5 peptides were culled, and the scattering profiles were computed for N130 pentamers, whilst excluding the rpL5 peptides. The N130 molecules include the oligomerization domains and disordered regions (see Fig. 1a in the main text). Details of how the scattering profiles were computed are as described in the methods section.

Comment 5: *Regarding the SANS data in Figure 2, the Authors state that in the A2 mutant a broad peak appears at intermediate q . However, the interpretation of this is a bit vague. It would be more cautious not to draw many conclusions from these SANS data as the measured intensity is a product of the form and structure factor of all components in the solution. The new variant (A2+) is not studied in detail, it is likely that the variant has a different form-factor compared to the WT. To address this aspect, SAXS/SANS experiments should be performed on both the WT and A2 mutant in order to obtain the experimental form-factor.*

Response to comment 5: We agree that it is important to be cautious about interpreting small shifts in the SANS curve. We have deleted mentions of the broad peak from the figure caption and rewritten the presentation of the results as follows (see text starting on line 232 of the revised manuscript):

Next, we investigated the impact of the +A2 mutant using SANS (Fig. 2e). We observed similar pairs of peaks at intermediate q -values for both N130 + rpL5 and the +A2 mutant + rpL5. Small shifts in the locations of the peaks are likely a combination of inherent noise and a contribution from electrostatic repulsions in the disordered N- and C-termini of N130 emanating from the same face of the PD⁷⁷. The C-terminus of the wild-type protein contains nine negatively charged residues corresponding to A2, and the +A2 mutant increases the net charge on the pentamer by 25..

Comment 6: *A quantitative evaluation of the peak position is also extremely relevant in relation to the data in Figure 3b. Indeed, the peak assignment from the simulations (53 Å, 95 Å and 140 Å) is considered in agreement with the SANS data (see caption of Figure 3b). However, between simulations and experiments there is a significant difference at least for two peaks (SANS peak estimation: 55, 77, 119 Å). Moreover, in the main text (page 9, first line) values of 52 Å, 100 Å and 145 Å are reported for what seems to be the analysis of simulations data in Figure 3b. If this is the case, could the Authors explain the mismatch between the peak positions from simulations reported in the main text and the ones reported in the caption?*

Response to comment 6: We have addressed this issue in response to comment 4 made by Reviewer 1. We direct the reviewer to our response to this comment and to the text starting on line 300 of the revised manuscript.

Comment 7: *Regarding the FRAP data, it would be beneficial to elaborate on how the recovery time is obtained. From the experimental data, the curves for WT and the A2 variant largely overlap if one considers the error bars. The reported data in page 7 only show errors in the range of 4-8%. Could the Authors comment on the significance of the calculated values compared to experimental data shown?*

Response to comment 7: Please see the following discussion starting on line 239 of the revised manuscript.

We also measured the impact of the +A2 mutant on the internal dynamics of N130 + rpL5. For this, we performed measurements of fluorescence recovery after photobleaching (FRAP) on the condensates (Fig. 2f). The FRAP curve for N130 + rpL5 indicates dynamical exchange with the bulk solution with the recovery time constant being 53 ± 2 s. Increasing the total charge on N130 via the +A2 mutant decreases the overall extent of FRAP, resulting in a longer recovery time of 103 ± 8 s. Similarly, we observe that N130^{+A2} + rpL5 displays slower overall dynamics at shorter timescales, and the dynamics of the two systems approach one another at longer times. The average recovery times were obtained by fitting the data, for both constructs, to a single species model. This does ignore the prospect of there being an immobile fraction. However, since FRAP data are a convolution of contributions from physical crosslinks and molecular transport, we chose a parsimonious, single species model to avoid over-fitting and over-interpretations of the data.

Comment 8: *Page 7 second paragraph, here the mutant N130a+2 is introduced. The text states “we replaced A0 with the residues from, in reverse order, to increase linear charge density”. Shouldn't it be with residues from A2? In the method section it is also not stated precisely what this new variant of N130 contains.*

Response to comment 8: Please see Fig. 2c for details of the sequence of the A0 region in the +A2 mutant. The text has been revised to clarify this point. Please see the text starting on line 221, which we reproduce below:

Our predictions motivated the generation of a new mutant construct where we replaced A0 with the residues from A2, in reverse order, to increase the linear charge density (see the sequence of the new A0 region in Fig. 2c). We refer to this construct as N130^{+A2}. It has more acidic residues than the wild type.

Comment 9: *For the SANS data of the variants in Figure 5 it would be useful to compare the effects the mutations have on the form-factor of the rpL5 variants (i.e., adding the experiments in the absence of N130). This would support the statement of the Authors that the changes in low-q are due to changes in the shape of the protein. This could be achieved by measuring SAXS or SANS of just the variants at different concentrations to obtain the experimental form factor. It would also be useful to have a proper analysis of the peak positions. The second peak at around 0.13 angstrom seems to initially move to slightly higher-q with increasing L but then for 20L it shifts to lower q. This combined with the drastic change at low q, observed for 20L, might suggest that the structure-factor is the measured scattering pattern, which could potentially cause the shift of the peak position at high-q combined with the broadening at low-q.*

Response to comment 9: We would like to collect more data for the linker variants. Unfortunately, the neutron source is down for upgrades, and will not be back online for the next 9-12 months. Therefore, we have deleted Figure 5, and we will save our analysis of these and other variants for a later date.

Comment 10: *In Figure 5, we can observe quite significant differences in the signal-to-noise ratio (especially at high q) of the SANS data (i.e., please compare the error on 10 L with 20L in figure 5a). Because it depicts a comparison with the SANS data in figure 1, one assumes that the same concentrations were used (of both proteins). However, this is not clear from either the caption, text or methods. If the same concentrations were used perhaps there was a different exposure time between the samples? Overall, it is hard to make a comparison between the data in this figure. Showing the data with an offset in the y-axis between the samples may clarify the statements about the data made.*

Response to comment 10: Please see our response to comment 9.

Comment 11: *Regarding the data in Figure 8. The Authors state that “The sub-diffusive dynamics derived from simulations are consistent with the apparent plateau in the FRAP data for the wild type (Fig. 2d) at long timescales”. Could the Authors elaborate more on this correlation between simulations and experiments? This is also related to our comment 7.*

Response to comment 11: Please see our response to comment 5 of Reviewer 1.

Responses to comments of Reviewer 3

Summary comments: *The paper by Dar et al provides an experimental and computational study of the multiscale structure of biomolecular condensates. Using in vitro experiments of N130+rpL5 condensates (that mimic nucleolar granular components) and computer simulations the authors show that these biomolecular condensates are network fluids with spatially inhomogeneous structure across different scales. Overall, this is a very interesting and well-written paper that contributes new insights into the emerging picture that biomolecular condensates are not spatially homogeneous liquids but rather viscoelastic materials with network-like organisation. I recommend publication in Nature Communications. I have a few minor comments, which I hope the authors will find useful in revising the manuscript:*

Response to summary comments: We thank the reviewer for this positive evaluation and for insightful comments to which we have provided point-by-point responses.

Comment 1: *In Fig 7, the authors show that N130+rpL5 condensates have bimodal degree distributions and hence show coexistence of gas- and liquid-like organization. How does one interpret the freely diffusing gas phase? Would this be similar to the dilute phase (which is “equilibrated” across the condensate interface)?*

Response to comment 1: As noted on line 408 of the revised manuscript: “**Here, the term “gas” refers to the presence of unassociated, freely diffusing rpL5 that coexists with a liquid comprising associated rpL5 molecules.**” It is worth noting that condensates are not polymer melts. Therefore, the gas-like regions are solvent-filled. We make this point starting on line 473 of the revised manuscript. This, we reproduce below:

We find that there are two types of sub-graphs that underlie the structure of the N130 + rpL5 condensates. One of the sub-graphs corresponds to gas-like organization, and the other corresponds to that of a liquid. Note that “gas-like” implies that there are regions within condensates where the concentrations of macromolecules are ultra-dilute, and hence solvent filled. This is akin to the empty liquid concept¹¹⁴ reported for patchy colloids. Conversely, what we refer to as “liquid-like” refers to regions that are dense in macromolecules. The bipartite graphs also have dynamical fingerprints, which are manifest as the bimodality we observe for the MSDs of the PDs. Super- and sub-diffusive behaviors that we report here have been observed in MSDs

computed from simulations of oligomer-grafted nanoparticles ¹¹⁵. They are also consistent with data from nuclear magnetic resonance experiments where Gibbs et al., found that the PDs of NPM1 form an immobilized scaffold in NPM1+p14ARF mixtures ¹¹⁶. Taken together, our findings place the N130 + rpL5 system, and other such systems, in the same category as patchy and / or hairy colloids ^{93, 94, 95, 97, 114, 115, 117, 118}.

Comment 2: *What determines the timescale t_D separating the super-diffusive and sub-diffusive regimes in the network fluids?*

Response to comment 2: As noted in the text, it is the interplay between the repulsions, which enhance transport, and attractive interactions, which generate physical crosslinks, that determine the timescale. The following text, please see line 426, clarifies these points:

Based on the observed length scales, the super-diffusive motion reflects the contributions of short-range steric repulsions among the PDs and the electrostatic repulsions between acidic residues. Conversely, the sub-diffusive motions reflect contributions from physical crosslinks between acidic residues and rpL5 peptides.

Comment 3: *In the conclusions, it would be interesting if the authors could expand the discussion on the nucleolus as a multiphase condensate. How do the authors expect the network structure to vary for the various phases in the nucleolus?*

Response to comment 3: We appreciate this request. However, given what we are learning about the GC, and the contributions made by Politz and Pederson, which have been ignored by those in the condensate literature who focus exclusively on NPM1, we think it would be a stretch to extrapolate from the current model system to say much that would stand up to scrutiny regarding nucleolar sub-phases or even GCs.

REVIEWERS' COMMENTS

Reviewer #1 (Remarks to the Author):

We are satisfied with the explanations and the changes made by the authors in response to the reviewers.

Reviewer #2 (Remarks to the Author):

We thank the Authors for the effort in addressing our points and for revising the manuscript, which is now, in our opinion, suitable for publication in Nature Communications.

Reviewer #3 (Remarks to the Author):

The authors have addressed all my comments. I have no further comments.